# Decoding Near Synonyms in Pedestrianization Research: A Numerical Analysis and Summative Approach

**Hisham Abusaada** [1,*] and **Abeer Elshater** [2]

1 Department of Architecture, Housing and Building National Research Center (HBRC), Cairo 1770, Egypt
2 Department of Urban Design and Planning, Ain Shams University, Cairo 1770, Egypt; abeer.elshater@eng.asu.edu.eg
* Correspondence: habusaada@yahoo.com

**Abstract:** Pedestrianization is a significant discourse focus within urban planning and design research. However, the need for more clarity from the inconsistent use of near-synonym concepts or terms necessitates attention. This review article addresses this issue through a comprehensive analysis of synonym proliferation in pedestrian research, culminating in developing a robust "near synonymous toolkit" and "synonym selection framework". Employing a linear snowball sampling technique, numerical analysis, and a qualitative content analysis-based summative approach, we examined sixteen peer-reviewed articles from 11 scientific journals. Through systematic classification based on consistency and variability, the summative review identifies three primary groups of near synonyms: dominant and widely utilized conceptual or terminological near synonymy in pedestrianization in the urban planning and design literature, near synonyms directly associated with a pedestrian, pedestrianize, and those indirectly linked to another conceptual or terminological synonymy. Further analysis delves into the nature of near-synonym concepts or terms, revealing three discernible patterns: the use of distinct, precise concepts or terms with near-synonym meanings, similar concepts or terms conveying divergent meanings, and the juxtaposition of unrelated vocabulary lacking semantic resemblance. These insights illuminate semantic relationships within the studied vocabulary, underscoring the importance of addressing inconsistency for clarity, precision, and coherence in scientific discourse. By offering practical guidance through the proposed framework, this study empowers academic researchers to navigate synonym selection adeptly, thereby enhancing the caliber of scholarly writing in urban planning and design.

**Keywords:** history of urban thought; pedestrian; synonyms; near-synonym terminology in urban studies; near-synonym concepts in urban planning; semantic preference; urban history

## 1. Introduction

The consistent usage of synonyms in scientific manuscripts is pivotal for ensuring clarity and precision in communication across diverse contexts, including the humanities in language and literature [1,2], and the social sciences in linguistics and language [3–6], culture [1,7], and architecture and urban studies [8].

This article delves into near synonyms, exploring expressions with similar but not identical meanings [5,9]. For example, Wandl and Magoni (2017) [10] suggest that the terms "transitional landscapes" and "terrains vagues" can be considered near synonyms for "peri-urban areas". These terms refer to the areas between urban and rural environments undergoing a land use and development transition. Similarly, the "territory of borders" can also be associated with peri-urban areas, implying a boundary or interface between urban and non-urban spaces. Likewise, van Kamp, Leidelmeijer, Marsman, and Hollander (2003) [11] and Abusaada and Elshater (2021) [12] present concepts, such as livability, living quality, living environment, the quality of the place, residential perception and residential satisfaction, the evaluation of the residential and living environment, quality of life, and sustainability, often used as synonyms.

These nuanced differences allow for alternative expressions that may be more suitable in specific contexts, fostering effective research collaboration, facilitating knowledge dissemination, and enabling seamless comparison between studies [13–15], including urban planning and design studies [8,11,16,17].

Sometimes, for example, some researchers use 'pedestrian area', 'pedestrian zone', 'pedestrian plaza', 'pedestrian footpath', and 'pedestrian environment' [18,19] or 'pedestrianized area' or 'pedestrianized street' [20] as well as terms like 'car-free streets', 'car-free zones', and 'car-free urban living' [21] as near synonyms, sharing common meanings but offering slight variations in nuance.

Pedestrianization stands at the heart of urban planning and design research, captivating scholarly interest and sparking ongoing debates. For a century, it has played a pivotal role in enhancing walkability and pedestrian experiences on city streets [21–23]. However, despite the well-established conceptual framework surrounding pedestrianization, the scholarly literature often lacks robustness in near-synonym terms for associated concepts, leading to linguistic complexities that impede effective communication. The widespread use of near synonyms introduces ambiguity, redundancy, inconsistency, and a lack of standardization in scientific discourse [24]. This complexity poses challenges for specialists, scholars, and policymakers seeking to manage pedestrian–vehicle interactions in urban environments.

Despite the progress made in peer-reviewed articles, this issue of inconsistent conceptual or terminological synonymy usage persists and needs to be addressed. The research identified a gap wherein pedestrianization, a pivotal topic in urban planning and design research, is marked by a significant proliferation of conceptual near synonyms and terminologies. These terms encompass various aspects such as specific areas, segments, overarching urban paradigms, methods, techniques, and actionable strategies within urban planning and design. To the best of our knowledge, no study has systematically examined the inconsistency of near synonyms for concepts or terms in pedestrianization research. This inconsistency poses a challenge in developing a comprehensive framework to guide future research in this field. Further investigation is warranted to gain deeper insights into using concepts and terms in pedestrianization research and their inter-relationships.

Given these challenges, this study explores the inconsistent use of near synonyms for concepts or terms in pedestrianization research. It aims to address this issue through three primary research objectives to enhance communication and comprehension. Firstly, it investigates the inconsistent usage of near synonyms for concepts or terms in contemporary studies. Secondly, it explores the "synonym selection framework" to facilitate effective and consistent synonym usage in pedestrian research. Thirdly, it delves into the significance of maintaining vocabulary consistency and its pivotal role in fostering communication and comprehension within pedestrian studies.

The overuse or inconsistent application of near synonyms may need to be clarified for readers, disrupt the narrative flow, and detract from a study's central message. This article endeavors to unravel the underlying complexities inherent in this endeavor. To this end, this article posits two primary research questions:

- Which toolkits, outlined in the literature, can ensure the effective and consistent use of near-synonym terms in pedestrianization research?
- Why is maintaining consistency crucial?
- How can near-synonym terms be selected judiciously?

Utilizing a linear snowball sampling technique [25] and a qualitative content analysis-based summative approach by Hsieh et al. (2005) [26], this study aims to contribute to the development of a 'near synonym toolkit' and 'synonym selection framework'. This framework serves as a valuable resource for novice academic researchers, assisting them in the selection of appropriate conceptual or terminological near synonyms related to pedestrianization for incorporation into their scholarly articles. The significance of this research lies in its thorough documentation of the increasing prevalence of near synonyms observed across various concepts and terms in the pedestrianization literature. This study evaluates

the relevance and thematic categorizations within the research domain, acknowledging the challenges posed by the abundance of near synonyms in the scholarly literature and emphasizing the importance of consistency, semantic preference, and systematic adherence.

Moreover, this study contributes by meticulously identifying the most consistent near synonyms within the specialized urban planning and design fields. This discernment adds value by fostering the construction of a knowledge repository that can steer practitioners towards more fitting solutions for urban challenges. Additionally, it furnishes academics with insights into handling near synonyms judiciously in forthcoming research endeavors. Addressing these challenges enriches the discourse on conceptual and terminological aspects of pedestrianization research writing, providing practitioners and academics in urban planning and design with a toolkit and framework for selecting near-synonymous concepts or terms.

## 2. Theory: The Near-Synonym Concepts or Terms in Pedestrianization Research

The proliferation of near-synonym terms and concepts poses challenges to standardization and clarity, highlighting the dynamic and evolving nature of academic discourse on pedestrianization. This review aims to provide a reliable theoretical background on the near synonym toolkit in the pedestrianization literature. By carefully embracing this diversity of near synonyms and selecting appropriate terms, researchers and practitioners can enrich their understanding of pedestrianization and contribute to a more comprehensive approach to urban planning and design.

The use of near-synonym concepts or terms in pedestrianization research within urban planning has been a recurring focus in previous studies, as demonstrated by the works of Monheim (1986) [27], Hass-Klau (2015) [28], and Robertson (1993) [29]. Monheim's (1986) [27] classification of pedestrian areas into distinct types based on their vehicular access restrictions sheds light on the varying degrees of pedestrianization implemented in different urban contexts. He introduced terms such as "pedestrianized central shopping streets" to describe specific locations where vehicular access was restricted, further illustrating the evolution of pedestrianization strategies over time. He delved in his work into the concept of "pedestrianization" within German cities, defining it as the closure of main shopping streets to vehicles and their subsequent redesign to align with newly constructed shopping centers. Moreover, he emphasized creating car-free environments to attract shoppers, viewing pedestrianization as a form of artistic urban renewal that fosters vibrant city centers.

Similarly, Robertson (1993) [29] discussed the concept of "pedestrian malls" in the United States, with examples such as Kalamazoo, Michigan, implementing downtown pedestrian malls as early as 1957. These malls aimed to provide pedestrians with a safe and inviting environment by separating them from vehicular traffic, akin to suburban shopping centers. The focus on creating pedestrian-friendly spaces underscores the importance of prioritizing pedestrian safety and accessibility in urban design. Hass-Klau (2015) [28] and Robertson (1993) [29] also employed the term "pedestrianization scheme" to describe the transformation of European city centers into pedestrian-only zones during the 1960s, highlighting the efforts to separate pedestrians from vehicular traffic.

Previously, scholars like Sadik-Khan (2001) [30], Gregg (2023) [22], and Brownrigg-Gleeson (2023) [31] have continued to explore the theme of pedestrianization, particularly in transforming urban spaces like Times Square and Broadway in Manhattan into pedestrian-friendly zones. They highlight the benefits of such schemes in improving pedestrian safety and enhancing the overall urban environment. These sentiments are like those expressed in earlier studies.

In their research paper, referencing Hass-Klau (2015) [28], Parajuli and Pojani (2018) [32] allude to Colin Buchanan's manuscript, "Traffic in Towns" (1987) [33], wherein he advocated for car-free zones as a potential solution to the conflict between pedestrians and automobiles. However, Parajuli and Pojani (2018) [32] utilized terms like "car-free streets", "car-free zones", and "car-free urban living" interchangeably to convey the same concept.

Moreover, introducing the concept of "complete streets" in 2003 represents a significant evolution in urban planning, aiming to accommodate various modes of transportation while prioritizing pedestrian safety and accessibility [34]. This concept builds upon previous ideas such as "pedestrianization" and "walkable streets", emphasizing inclusivity and sustainability in urban design. The widespread adoption of complete street principles across the United States and Canada reflects ongoing efforts to create safer and more inclusive urban environments for all residents.

Robertson (1993) [29] sees 'pedestrian malls' as a form of pedestrianization that originated in the United States, with Kalamazoo, Michigan, in the USA, being the first city to implement a downtown pedestrian mall in 1957. These malls are designed to provide a comfortable environment for pedestrians by separating them from automobile traffic. Like suburban shopping centers, pedestrian malls offer convenience, accessibility, and safety to shoppers and visitors. They are often characterized by beautifully landscaped park-like corridors in the town center, discouraging automobile traffic and creating a welcoming atmosphere.

Like the near synonym similarity in previous studies, Janette Sadik-Khan (2017) [30] and Kelly Gregg (2023) [22] believe that Times Square, portions of Broadway in Midtown Manhattan, and numerous pedestrian plazas throughout the borough were transformed into pedestrian malls. While Brownrigg-Gleeson (2023) [31] used it to suggest that such schemes were designed to improve pedestrian safety and create more pedestrian-friendly urban environments, he gave the example of Stroget Street, Copenhagen, in 1962, noting that a positive outcome of pedestrianization is pedestrian prioritization.

Throughout history, a similar trend has been observed with the concept and term "complete street", which emerged in 2003 to improve the interaction between pedestrians and drivers while addressing the transportation needs of all road users, including pedestrians, cyclists, motorists, and public transportation users [34]. Initially introduced by David Goldberg in "*America with Smart Growth*", the slogan "Complete Streets" has gained prominence today [35].

This approach seems to have aspects in common with previous concepts such as "pedestrianization", "streets for pedestrians only", "streets for people", and "walkable streets" by emphasizing the importance of accommodating all forms of transportation. Complete streets aim to create a safe, sustainable, and inclusive environment. It is also known as a term and a concept. Its principles, prioritizing pedestrians, cyclists, and transit users, have been applied in many cities across the United States and Canada [4,34].

As such, this review of the above literature can be a valuable resource for improving the clarity, precision, and effectiveness of scholarly writing, particularly in urban planning and design.

## 3. Methodology

Our research methodology employs both quantitative and qualitative research techniques, providing a comprehensive understanding of the research topic. Utilizing the linear snowball sampling technique, we curated a selection of sixteen peer-reviewed articles from eleven esteemed scientific journals relevant to our research domain. To ensure the alignment with the research objectives to ensure the validity and reliability of the findings, the criteria of selection include the following [36–38]:

–   Relevance to pedestrianization research: Articles should be relevant to the research topic or area of pedestrianization being investigated. This ensures that the analysis uses the near-synonym concepts or terms meaningfully.
–   Publication Date: The researchers of the current study utilized the selected articles published within the last ten years (2014–2024).
–   Publication Type: Our bibliometric analyses focused on scholarly articles published in peer-reviewed journals.
–   Language: The language of publications may be a criterion depending on the scope of the study and the language proficiency of the researchers of the current work

to facilitate conducting the analysis. We focused only on articles published in the English language.

- Source: We selected journals indexed in SCImago, which provided a platform for articles in these journals that were indexed in the Scopus database. Choosing reputable sources such as Q1 and Q2 journals was essential to ensure the data's quality and reliability.

- Exclusion criteria: We also defined exclusion criteria to filter out irrelevant articles or those not meeting specific quality standards. Accordingly, we excluded books, book sessions, and conference proceedings. This may include excluding articles published in predatory journals or indexed journals that are among the lowest quartiles (Q3 and Q4 journals) and focusing on Q1 and Q2 journals only.

Through quantitative and qualitative research techniques, we conducted a qualitative content analysis-based summative examination to unearth fundamental meanings and significance within the literature. This method systematically analyzes qualitative data to uncover various near synonyms related to specific concepts or terms, focusing on identifying and exploring different words or phrases conveying similar meanings. Subsequently, the key findings were identified and summarized cohesively. The research design unfolded through eight distinct steps.

First, we employed a linear snowball sampling technique to identify relevant journals. This technique involved progressively identifying seminal articles or sources related to "pedestrianization" and scrutinizing the cited references to expand our source pool. The initial stage culminated in identifying a substantial volume of references that fell outside our inclusion criteria. We purposefully excluded gray literature sources, such as book series, conference proceedings, trade journals, and non-peer-reviewed publications. We also excluded sources lacking the primary search term "pedestrianization" in their titles.

Secondly, we relied on the assumption that relevant articles would include additional keywords in their abstracts or the main text of the selected papers. Through iterative searches and an exploration of the existing literature, we identified a set of keywords relevant to our research, including "pedestrian", "pedestrianization", "pedestrianized streets", "pedestrian areas", "pedestrian zones", "pedestrian malls", and "for pedestrian-only". Our search query combined these keywords using logical operators such as "AND" and "OR" to ensure comprehensive coverage. The finalized search query looked like this: "pedestrian" AND "pedestrianization" OR "pedestrianized streets" AND "pedestrian areas" OR "pedestrian zones" AND "pedestrian malls" OR "shopping malls" OR "shopping centers" AND "for pedestrian-only" AND "pedestrian friendly".

Third, our investigation was meticulously designed to encompass various disciplines and researchers' specializations. We aimed to investigate these diverse areas to elucidate their intricate interactions and implications for the diversity of near synonyms. Therefore, we meticulously scrutinized journals containing the initially selected articles based on various criteria, including journal subject areas and categories, *h*-index values, and quartiles (Q1 and Q2), leveraging data from 2024. Thus, we identified sixteen published articles. Further details are available in Figure 1 and Table 1 (see also Table S1 in Supplementary Material), which includes sixteen articles published between 2014 and 2024.

Fourth, we commenced by examining how the concept of pedestrianization has been discussed in the sixteen selected articles over the span of a decade. Fifth, we rigorously assessed the relevance and applicability of identified sources to our research objectives using the qualitative content analysis-based summative approach. A numerical analysis was conducted to ascertain the frequency of keywords across the literature. Then, we examined how these keywords related to other terms that consistently appeared alongside them in titles, abstracts, and index words. Through this analysis, we identified near-synonym concepts or terms within the field of study. Details of this analysis, including numerical findings, are provided in Supplementary Material (File S1).

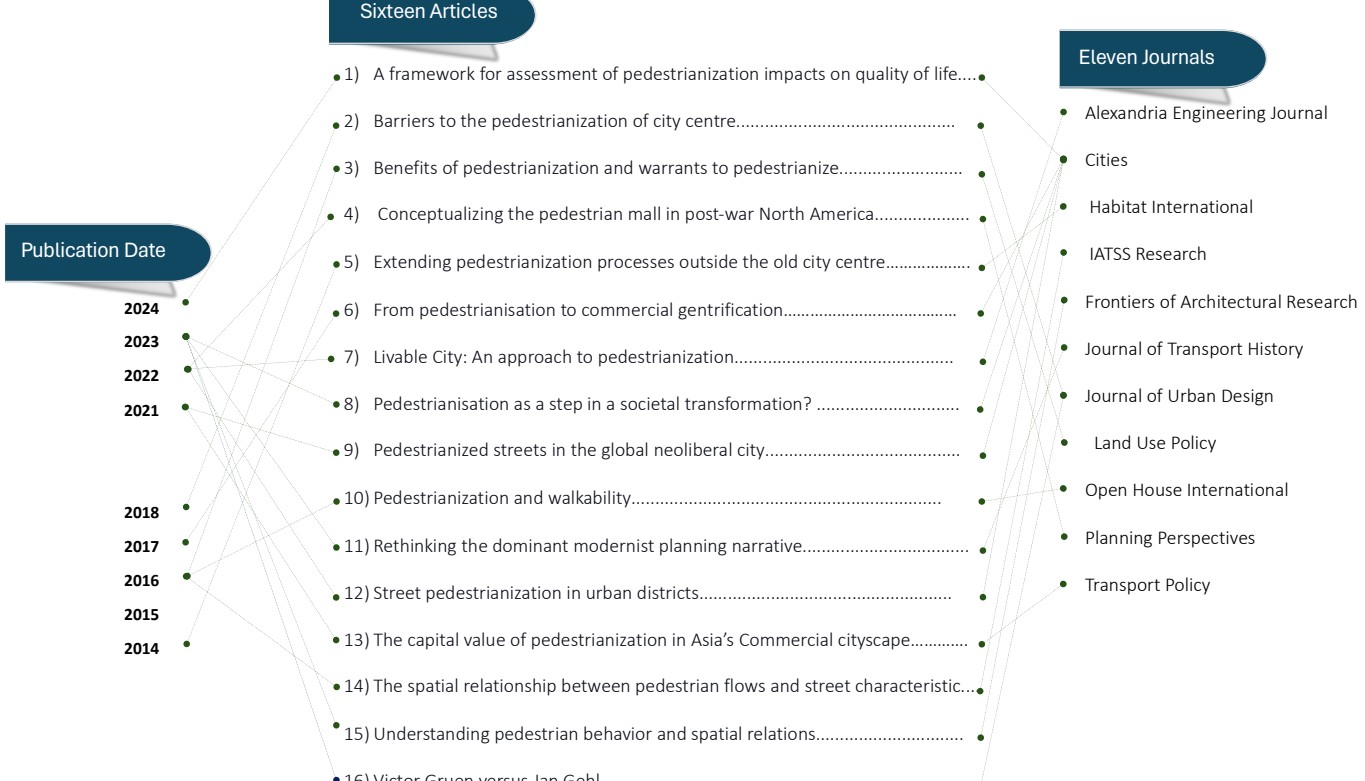

**Figure 1.** Eleven journals holding 16 articles are listed in the SCImago database between 2014 and 2024 [3,21,22,32,39–50].

Sixth, we reviewed the text to identify frequently used concepts and distinguish between independent and near-synonym terms. This process aimed to clarify the usage and relationship of these terms within the context of pedestrianization research. Seventh, the qualitative content analysis-based summative approach results were systematically classified into three distinct groups, delineating dominant near synonyms, terms related to pedestrian concepts, and indirectly related near synonyms. These classifications provided valuable insights into the consistency and diversity of near-synonym terms that authors use.

Eighth, drawing on these findings, the discussion aimed to elucidate the nature of near-synonym terms and identify distinct patterns in their usage. We categorized near synonyms into patterns through meticulous analysis, facilitating a deeper understanding of semantic relationships and nuances within the investigated vocabulary.

In our effort to examine consistency in pedestrianization research regarding conceptual or terminological synonymy, we developed a comprehensive four-step strategy for analyzing the main text of selected articles. Firstly, we employ an analytical approach to distinguish between distinct and independent concepts or terms and their corresponding near synonyms identified in previous research. Secondly, we concentrate on concepts or terms directly pertinent to the study's subject matter, avoiding unnecessary digressions. Thirdly, we meticulously scrutinize the text to identify and eliminate near synonyms with multiple meanings or interpretations. Finally, we ensure that the concepts or terms introduced in the research are thoroughly defined and consistently used throughout the text while monitoring for repetition.

**Table 1.** Overview of journals and articles in SCImago and Scopus databases published between 2014 and 2024.

| Author(s) | Specialization | Journal | Subject Area | Categories | *h*-Index * | Q ** |
|---|---|---|---|---|---|---|
| Allirani et al. (2024) [3] | Civil engineering | | | | | |
| te Boveldt, Wilde, Keseru, and Macharis (2023) [39] | Sustainable and urban mobility and spatial planning | | | | 114 | 1 |
| Yoshimura et al. (2022) [40] | Advanced science and technology, urban studies, and planning | *Cities* | | | | |
| Villani and Talamini (2021) [51] | Architecture and civil engineering | | | Development, sociology and political science, and urban studies | | |
| Özdemir and Selçuk (2017) [41] | Architecture | | Social sciences | | | |
| Murakami, Villani, and Talamini (2021) [42] | Architecture and civil engineering, and arts and social sciences cluster | *Transport Policy* | | Development, sociology and political science, and urban studies | 113 | 1 |
| Gregg (2019) [43] | Geography and planning | *Planning Perspectives* | | | 31 | 2 |
| Zainol et al. (2016) [44] | Built environment | *Open House International* | | | 15 | 2 |
| Soni and Soni (2016) [45] | Urban transport planner; civil engineering | *Land Use Policy* | | | 138 | 1 |
| Gregg (2023) [22] | Geography and Planning | *Journal of Urban Design* | | | 54 | 1 |
| Parajuli and Pojani (2018) [32] | Earth and environmental sciences | | | | | |
| Nakamura (2016) [46] | Urban studies, urban design, and planning | *IATSS Research* | | Urban studies | 36 | 1 |
| Castillo-Manzano, Lopez-Valpuesta, and Asencio-Flores (2014) [47] | Applied economics and management research group | *Habitat International* | | Urban studies | 102 | 1 |
| Feriel (2023) [21] | History | *Journal of Transport History* | Arts and humanities | History | 19 | 1 |
| Yıldırım and Çelik (2023) [48] | Architecture | *Frontiers of Architectural Research* | Engineering | Architecture building and construction | 35 | 1 |
| Yassin (2019) [49] | | *Alexandria Engineering Journal* | | Engineering (miscellaneous) | 81 | 1 |

* '*h*-index' means measuring the research impact, and ** Q means the best quartile, which is the ranking of a journal based on the database of SCImago Journal & Country Rank (SJR).

## 4. Results

The interplay between concepts and terms is significant in language and communication [52,53]. A concept is an abstract idea that helps understand and categorize the world, while a term is a specific expression that represents a concept more concretely. Thus, a concept is a mental tool for analysis, while a term is a linguistic tool used to describe and identify things. Multiple terms can represent distinct aspects or forms of the same concept. Figure 2 shows the structure of the research.

### 4.1. Decadal Discourse: Conceptualizing Pedestrianization

Over the past decade, urban planning and design scholars have characterized pedestrianization in multifaceted ways. It has been conceptualized as a process [41,47], a measure, a device, key components [32], and a strategy [3,43] or policy [40,45,51]. Additionally, it has been conceptualized as an approach [22,45,48,50] or a pivotal step in societal transformation [39] and an international phenomenon [21]. This decadal discourse indicates its multifaceted nature and significance in urban development.

The term "pedestrianization" often implies the allocation of space for pedestrians; however, scholars have delineated it further, distinguishing it from concepts like "traffic calming" and emphasizing its social, economic, and environmental implications [32,41,47].

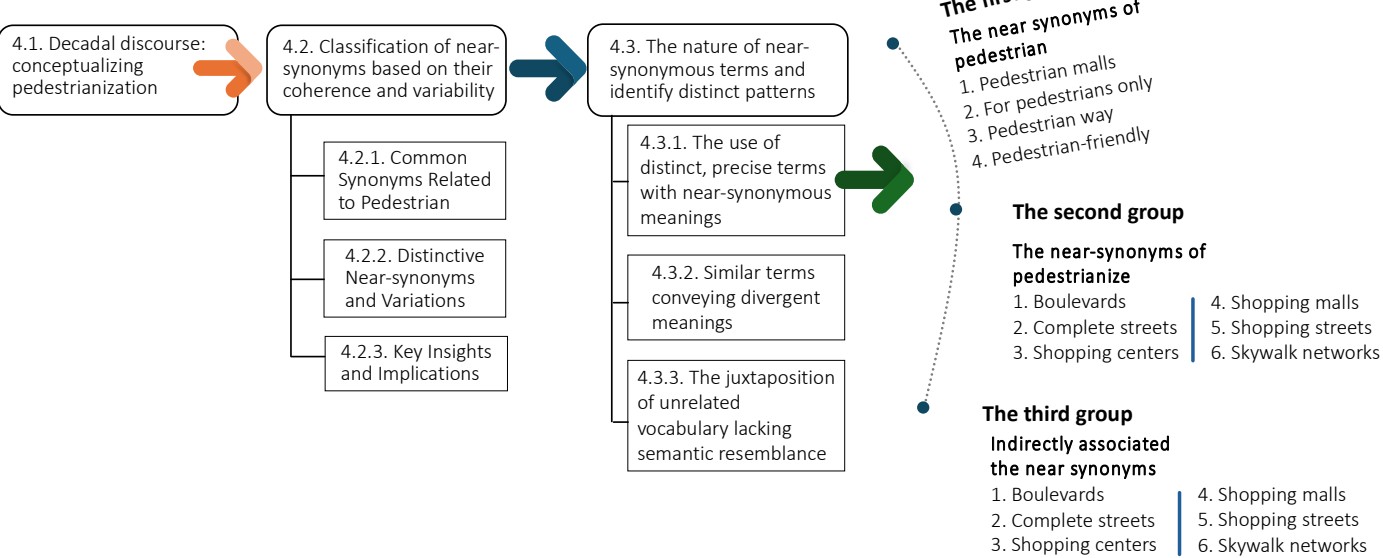

**Figure 2.** The structure of the current results. Three groups of near synonyms (see Table 2).

**Table 2.** The numerical analysis of near-synonym concepts and terms derived from the examination of sixteen articles categorized into three groups. The numbers enclosed in brackets represent the count of occurrences of near synonyms found within each examined manuscript.

| No. | Author(s) | Dominant Near Synonyms | Correlated Near Synonyms Related to Pedestrianization | Differentiated Near Synonyms |
|---|---|---|---|---|
| 1. | Allirani et al. (2024) [3] | Pedestrian/pedestrians (42), pedestrian mall/malls (2), pedestrian street/streets (7), pedestrian zone/zones (5), pedestrian-friendly footpaths (1), pedestrian footfall (7) | Pedestrianization (76), pedestrianized (29), pedestrianized urban street/streets (5) | Walkability (12), walking environment (1) |
| 2. | Feriel (2023) [21] | Pedestrian/pedestrians (56), pedestrian mall/malls (3), pedestrian area/areas (5), pedestrian street/streets (4), pedestrian zones (1), pedestrian schemes (2), pedestrian-oriented (2), shopping streets (5) | Pedestrianization (44), pedestrianized (7), pedestrianized area (2) | Car-oriented (2), car-free (8), car-free zones (2), traffic-free areas (1), walkability (5), walkable city (6), walkable cities (1) |
| 3. | te Boveldt et al. (2023) [39] | Pedestrian/pedestrians (13), pedestrian area/areas (5) | Pedestrianization (83), pedestrianization scheme (12), pedestrianized zone (1), pedestrianized center (1), pedestrianizing (5), pedestrianizing boulevard (1), pedestrianizing central boulevards (1) | Boulevards (9), car-free boulevard (4) |
| 4. | Gregg (2023) [22] | Pedestrian/pedestrians (117), pedestrian mall/malls (45), pedestrian streets (4), pedestrian strategies (1), pedestrian mall strategies (2), pedestrian street design (2) | Pedestrianization (84), pedestrianization strategies (16), post-war pedestrianization (5), pedestrianizing streets (2) | Complete streets (1) |
| 5. | Yıldırım and Çelik (2023) [48] | Pedestrian/pedestrians (166), pedestrian zones (1), pedestrian-friendly (1), pedestrian behavior (54), pedestrian mobility (1) | Pedestrianization (39), pedestrianization policy (2), street pedestrianization (2), pedestrianized (9), pedestrianized area/areas (17), pedestrianized place (1) | Pedestrian-oriented alternatives (1) |

**Table 2.** *Cont.*

| No. | Author(s) | Dominant Near Synonyms | Correlated Near Synonyms Related to Pedestrianization | Differentiated Near Synonyms |
|---|---|---|---|---|
| 6. | Yoshimura, et al. (2022) [40] | Pedestrian/pedestrians (56), pedestrian area/areas (5), pedestrian environments (6), pedestrian space (5), pedestrian policies (1), pedestrian-friendly environment (1), pedestrian-friendly spaces (1) | Pedestrianization (39), pedestrianized (9), street pedestrianization (2), pedestrianized place (1), pedestrianization policy (2) | Vehicle-oriented (1), walkable environment (1) |
| 7. | Villani and Talamini, (2021) [51] | Pedestrian/pedestrians (39), pedestrian schemes/schemes (4) | Pedestrianization (93), pedestrianized (58), pedestrianized streets (10), pedestrianized area (15), pedestrianized space (11) | Walkable city (2), walkable street (1) |
| 8. | Murakami et al. (2021) [42] | Pedestrian/pedestrians (65), pedestrian malls (3), pedestrian zone (11), pedestrian zone schemes (26), pedestrian-only zones (1), pedestrian-friendly design (1) | Pedestrianization (36), pedestrianized (1), pedestrianization (4), pedestrianization schemes (1) | Skywalk networks (45), walkable built environments (4), transit-oriented locations (1), traffic-calming (1), car-dependent suburbia (2) |
| 9. | Gregg (2019) [43] | Pedestrian/pedestrians (10), pedestrian mall/malls (129), downtown pedestrian malls (128), pedestrian area/areas (11), pedestrian zone/zones (5), pedestrian space (1), pedestrian plaza (1), pedestrian way (4), downtown pedestrian streets (1), for pedestrians only (3), shopping centers (3), suburban shopping centers (17) | Pedestrianization (57), pedestrianization schemes (2), pedestrianized (13), pedestrianized streets (5), pedestrianizing (5), pedestrianizing main streets (2) | Pedestrian-oriented suburban post-war suburban shopping centers (1) |
| 10. | Yassin (2019) [49] | Pedestrian/pedestrians (38), pedestrian area/areas (3), pedestrian place/places (1), pedestrian priority scheme (1), pedestrian-oriented scheme (1), | Pedestrianization (61), pedestrianization schemes (9), pedestrianized (2), | Walkability (1), walkable environment (4) |
| 11. | Parajuli and Pojani (2018) [32] | Pedestrian/pedestrians (124), pedestrian mall/malls (81), pedestrian area/areas (2), pedestrian street/streets (4), pedestrian outdoor streets (1), pedestrian-oriented (1) | Pedestrianization (71), pedestrianization schemes (7), pedestrianized (9), pedestrianized area (2), non-pedestrianized) streets (1), pedestrianized districts (1), pedestrianized city centers (1) | Car-free streets (1), car-free zones (1), car-free urban living (1), car-oriented city (1), walking cities (1) |
| 12. | Özdemir and Selçuk (2017) [41] | Pedestrian/pedestrians (50), pedestrian area/areas (1), pedestrian zone/zones (1), pedestrian environments (1), pedestrian pockets (1), shopping streets (3), shopping malls (1) | Pedestrianization (81), pedestrianization scheme (29), pedestrianized (24), pedestrianized areas (24) | Walkability (1), un-walkable (1) |
| 13. | Zainol, et al. (2016) [44] | Pedestrian/pedestrians (33), pedestrian way (2), pedestrian walkway (7), pedestrian zone/zones (1) | Pedestrianization/ pedestrianizations (21), pedestrianizing areas (1) | Walkability (83), walkable environment (4), walkable street corridor (1), walkability friendly environment (2) |
| 14. | Nakamura, (2016) [46] | Pedestrian/pedestrians (31), pedestrianized space/spaces, (2), pedestrian-friendly environments (1), pedestrian-friendly built environments (1) | Pedestrianization (7), pedestrianize (70), pedestrianized area/areas (2), pedestrianized street/streets (36), non-pedestrianized (19) | Traffic-calming (6) |
| 15. | Soni and Soni (2016) [45] | Pedestrian/pedestrians (32), pedestrian area/areas (15), pedestrian street/streets (3), for pedestrians only (1), pedestrian friendly (4), pedestrian-friendly environment (1), pedestrian-friendly schemes (1) | Pedestrianization (104), pedestrianization schemes (12), pedestrianization of an area (2), pedestrianize (4), pedestrianized (37), pedestrianized area/areas (6), pedestrianized streets (1) | Car-friendly land use (1), underground pedestrian corridors (1) |

**Table 2.** *Cont.*

| No. | Author(s) | Dominant Near Synonyms | Correlated Near Synonyms Related to Pedestrianization | Differentiated Near Synonyms |
|---|---|---|---|---|
| 16. | Castillo-Manzano, et al. (2014) [47] | Pedestrian/pedestrians (30), pedestrian zone/zones (19), pedestrian street/streets (4) | Pedestrianization (1), pedestrianizing areas (1), pre-pedestrianization (1), pedestrianization schemes (7), pedestrianization processes (14), pedestrianized (18), pedestrianized street/streets (8), pedestrianized zone/zones (4) | - |

Castillo-Manzano et al. (2014) [47] referred to it as "a pedestrianization process", distinguishing it from "traffic calming", which relies on physical measures to treat the road, such as road bumps, tree plantings, or speed bumps, to convince the driver that the street is primarily intended for shopping or residential use. This process also fundamentally impacts the urban quality of the environment, emphasizing that it is a process that no longer adheres to city centers but instead expands them outside those centers. They gave an example of the city of Seville, Spain. In the same vein, Castillo-Manzano et al. (2014) [47] and Özdemir and Selçuk (2017) [41] returned to point out that it is a process that has social consequences that lead to population satisfaction and increase visitation rates to the converted areas, as well as that its goal is to improve the local physical environment and increase economic activity. Özdemir and Selçuk (2017) [41] gave an example from Istanbul, the Asian side of Kadıköy's historic center and retail area.

Parajuli and Pojani (2018) [32] characterize pedestrianization as a set of measures designed to address various environmental challenges, including gaseous, sound, and visual pollution. They emphasize its role in fostering positive social impacts by encouraging pedestrian activity, facilitating social interactions, and promoting tourism. From an economic standpoint, pedestrianization is viewed as a strategy to ensure the financial viability of retail establishments, particularly in the face of increasing competition from suburban shopping centers. The authors highlight transforming city centers into pedestrian-friendly spaces to enhance these areas' attractiveness and promote economic development. They underscore pedestrianization as a critical component of broader urban development projects aimed at revitalizing city centers and creating vibrant, livable communities.

Gregg (2019) [43] views pedestrianization as a holistic urban regeneration strategy to revitalize entire city centers, explicitly focusing on rejuvenating downtown retail areas by converting main streets into pedestrian zones. In a subsequent study, Gregg (2023) [22] reaffirms pedestrianization as a crucial urban planning and design approach. She advocates for converting streets into pedestrian malls as a strategic response to the decline in retail activity in city centers caused by the proliferation of suburban shopping centers, drawing inspiration from Victor Gruen's proposals. Additionally, she highlights pedestrianization and pedestrian malls as innovative strategies to enhance public spaces in urban environments. Similarly, Allirani et al. (2024) [3] recognize pedestrianization as a valuable strategy for mitigating the challenges of increased vehicular traffic in urban areas. They assert that pedestrianization positively impacts cities by improving overall livability and enhancing the quality of life, particularly from an economic perspective.

Soni and Soni (2016) [45] approach the concept as a policy directive to foster the development of pedestrian-friendly transportation systems, promote non-motorized transportation alternatives, and advocate for transit-friendly infrastructure and supportive land use patterns. This perspective aligns with the approach taken by Villani and Talamini (2021) [51] who delineate a policy framework comprising four key objectives tailored to Hong Kong's urban context. These objectives encompass ensuring pedestrian mobility and safety, integrating commercial functions within pedestrianized areas, enhancing mobility and accessibility around transport hubs, and conducting ongoing multidisciplinary studies about pedestrian planning. Likewise, studies by Soni and Soni (2016) [45] and Uzunoğlu

and Uzunoğlu (2020) [50] espouse the concept as an essential approach for establishing traffic-free zones and streets, particularly in historic urban centers where vehicular congestion poses significant challenges. Feriel's study (2023) [21] is deeply influenced by the seminal work of Breines and Dean, "The Pedestrian Revolution: Streets without Cars", published in 1974. In Feriel's analysis, "walking" takes center stage, underscoring the essence of pedestrian movement. He notably observes that limited pedestrianization is broached within the context of pedestrian streets, a topic not extensively explored since the 1950s [21].

*4.2. Classification of Near Synonyms Based on Their Coherence and Variability*

The Results section of this study investigates near synonyms associated with the concept and term 'pedestrianization'. Through a qualitative content analysis-based summative approach of the 16 articles, near synonyms of concepts or terms are systematically categorized into three groups: dominant near synonyms related to 'pedestrian', correlated near synonyms to pedestrianization (including 'pedestrianize', 'pedestrianizations', 'pedestrianized', and 'pedestrianizing'), and other related near-synonym concepts or terms, including walkability, and miscellaneous, such as boulevards, complete streets, car-free environment or space, skywalks, shopping areas or centers or malls or streets, and traffic-free areas or traffic-calming.

Figure 3 provides an overview of the coherence and diversity of concepts or near-synonym terms based on the analysis of the sixteen articles across these three groups. Table 2 presents a numerical analysis of concepts or terms extracted from a systematic examination (content analysis) of sixteen articles, categorized into three groups for clarity. The first group encompasses dominant and widely employed near synonyms found in the urban planning and design literature. The second group includes near synonyms directly relevant to the pedestrianization concept or term, along with those associated with other related concepts. The third group comprises distinct near synonyms indirectly related to pedestrianization. These visual and tabular representations offer valuable insights into the coherence and variability of near-synonym concepts or terms utilized by authors, facilitating the synthesis of findings and identification of key themes across the literature.

The previous table provides a comprehensive analysis of the near synonyms employed in relation to pedestrian-related concepts or terms, revealing both commonalities and distinctions among the works. Notably, the analysis highlights that some near synonyms are mentioned only once in the text, while others are scattered throughout the articles, conveying similar meanings but used sporadically. For instance, terms such as "pedestrian-friendly footpaths", "pedestrian schemes", "pedestrian-oriented", "pedestrian strategies", and "pedestrian mall strategies" are each employed by many authors only once, indicating a scattered use of near synonyms with similar meanings.

4.2.1. Common Synonyms Related to Pedestrian

Authors utilize a set of near synonyms commonly associated with pedestrianization. These include terms such as "pedestrian mall/malls", "pedestrian area/areas", "pedestrian zone/zones", and "pedestrian street/streets". Additionally, correlated near synonyms like "pedestrianized", "pedestrianized area", and "pedestrianization scheme" are prevalent across articles. Furthermore, terms related to "walkability" or "car-free", such as "walking environment", "walkable city", and "car-free zones", are consistently employed.

4.2.2. Distinctive near Synonyms and Variations

While there are commonalities in the use of pedestrian-related near synonyms, variations also exist among the authors' works. For instance, Feriel (2023) [21] demonstrates a broader range of near synonyms compared to other authors, incorporating terms like "pedestrian-oriented" and "pedestrian malls". Additionally, certain authors introduce unique terms such as "pedestrian footfall" and "post-war pedestrianization", highlighting their specific focus areas. Furthermore, variations are observed in the usage of terms related

to urban streets and boulevards, with some authors emphasizing concepts like "boulevards" and "shopping streets".

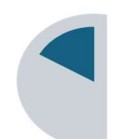

**Pedestrian**

1. Pedestrian areas
2. Pedestrian districts
3. Pedestrian environments
4. Pedestrian footfall
5. Pedestrian-friendly environment
6. Pedestrian-friendly built environments
7. Pedestrian-friendly schemes
8. Pedestrian-friendly spaces
9. Pedestrian malls
10. Pedestrian mall strategies
11. Pedestrian mobility
12. Pedestrian-only
13. Pedestrian-only streets
14. Pedestrian-only zones
15. Pedestrian-oriented
16. Pedestrian-oriented scheme
17. Pedestrian-oriented streets
18. Pedestrian outdoor streets
19. Pedestrian places
20. Pedestrian pockets
21. Pedestrian priority scheme
22. Pedestrian priority streets
23. Pedestrian schemes
24. Pedestrian streets
25. Pedestrian street design
26. Pedestrian  traffic
27. Pedestrian way
28. Pedestrian walkway
29. Pedestrian zones
30. Pedestrian zone schemes1.

**Pedestrianize**

**Pedestrianization**

1. Pedestrianization an area
2. Pedestrianization schemes
3. Pedestrianization strategies
4. Pedestrianization of the central boulevards
5. Pedestrianization process
6. Post-war pedestrianization
7. Pedestrianization policy
8. Pre-pedestrianization
9. Street pedestrianization

**Pedestrianized**

1. Pedestrianized area
2. Pedestrianized centre
3. Pedestrianized districts
4. Pedestrianized place
5. Pedestrianized space
6. Pedestrianized streets
7. Pedestrianized square
8. Pedestrianized zone
9. Pedestrianized urban street
10. Pedestrianized city centres
11. Non-pedestrianized

**Pedestrianizing**

1. Pedestrianizing area
2. Pedestrianizing boulevard
3. Pedestrianizing main streets
4. Pedestrianizing streets

**Other related synonyms concepts or term**

**Walkability**

1. Walkable built environments
2. Walkable environment
3. Walkability friendly environment
4. Walkable city
5. Walking cities
6. Walkable streets
7. Walkable street corridor
8. Walkway

**Miscellaneous**

1. Auto-restricted zones
2. Boulevards
3. Complete streets
4. Car-dependent suburbia
5. Car-free environment
6. Car-free space
7. Downtown pedestrian malls
8. Downtown pedestrian streets
9. Skywalks
10. Skywalk networks
11. Shopping areas
12. Shopping centers
14. Shopping malls
15. Shopping streets
16. Suburban shopping centre
17. Traffic-free areas
18. Traffic-calming

**Figure 3.** An overview of the coherence and diversity of concepts or near-synonym terms according to the analysis of the sixteen articles in three groups.

### 4.2.3. Key Insights and Implications

Understanding the nuanced usage of pedestrian-related near synonyms is essential for advancing scholarship and informing urban planning and design practices. By identifying both commonalities and distinctions in synonym usage, researchers can streamline terminology and enhance clarity in scholarly discourse. Future research could delve deeper into the implications of synonym usage patterns, aiming to foster more cohesive and effective communication within the field of pedestrianization research.

### 4.3. The Nature of Near-Synonym Terms or Concepts and Identify Distinct Patterns

Drawing on these findings, this subsection aims to elucidate the nature of near synonyms and identify three distinct patterns: the use of distinct, precise terms with near-synonym meanings, similar terms conveying divergent meanings, and the juxtaposition of unrelated vocabulary lacking semantic resemblance.

### 4.3.1. The Use of Distinct, Precise Terms with Near-Synonym Meanings

This subsection consolidates the findings into three distinct groups: pedestrian, pedestrianize, and indirectly associated near synonyms. These groups encapsulate various terms that are utilized interchangeably, each contributing to the discourse on pedestrianization. These groups represent a spectrum of linguistic nuances, each contributing uniquely to the discussion surrounding pedestrian-friendly urban spaces.

- The first group: The near synonyms of pedestrian

The initial cluster of near-synonym terms belongs to the first category and centers around the term "pedestrian", signifying movement on foot. This term is recurrent throughout the works of all authors to varying degrees. Notably, Gregg (2019) [43] employs it most frequently, while Yıldırım and Çelik (2023) [48] and te Boveldt et al. (2023) [39] use it less frequently.

The different terms used to refer to an area where pedestrians are given priority include pedestrian areas or areas, pedestrian zones or zones, and downtown. A diverse array of terms is deployed to delineate areas prioritizing pedestrian movement, ranging from "pedestrian areas" or "areas" and "pedestrian zones" or "zones" to "downtown pedestrian streets". Similarly, various appellations denote segments of urban landscapes designed for pedestrian usage, including "pedestrian mall/malls", "pedestrian ways", "pedestrian walkways", "pedestrian streets" or "streets", "pedestrian places" or "places", "pedestrian spaces", "pedestrian outdoor streets", and "pedestrian plaza" (Table 3).

**Table 3.** Overview of numerical analysis of near synonyms of pedestrians.

| Notes | Manuscripts * / Near Synonym | 1 | 2 | 3 | 4 | 5 | 6 | 7 | 8 | 9 | 10 | 11 | 12 | 13 | 14 | 15 | 16 |
|---|---|---|---|---|---|---|---|---|---|---|---|---|---|---|---|---|---|
| Urban segments | Pedestrian/pedestrians | 42 | 56 | 13 | 117 | 166 | 56 | 39 | 65 | 10 | 38 | 124 | 50 | 33 | 31 | 32 | 30 |
| | Pedestrian area/areas | - | 5 | 5 | - | - | 5 | - | - | 11 | 3 | 2 | 1 | - | - | 15 | - |
| | Pedestrian districts | - | - | - | - | - | - | - | - | - | - | - | - | - | - | - | - |
| | Pedestrian environments | - | - | - | - | - | 6 | - | - | - | - | - | 1 | - | - | - | - |
| | Pedestrian plaza | - | - | - | - | - | - | - | - | 1 | - | - | - | - | - | - | - |
| | Pedestrian zone/zones | - | 1 | - | - | 1 | - | - | 11 | 5 | - | - | 1 | 1 | - | - | 19 |
| Segments of urban landscape design | Pedestrian mall/malls | 2 | 3 | - | 45 | - | - | - | 3 | 129 | - | 124 | - | - | - | - | - |
| | Pedestrian outdoor streets | - | - | - | - | - | - | - | - | - | - | 1 | - | - | - | - | - |
| | Pedestrian place/places | - | - | - | - | - | - | - | - | - | 1 | - | - | - | - | - | - |
| | Pedestrian pocket/pockets | - | - | - | - | - | - | - | 1 | - | - | - | 3 | - | - | - | - |
| | Pedestrian space/spaces | - | - | - | - | - | 5 | - | - | 1 | - | - | - | - | 2 | - | - |
| | Pedestrian street/streets | 7 | 4 | - | 4 | - | - | - | - | - | - | 4 | - | - | - | 3 | 4 |
| | Pedestrian way | - | - | - | - | - | - | - | - | 4 | - | - | - | 2 | - | - | - |
| | Pedestrian walkway | - | - | - | - | - | - | - | - | - | - | - | - | 7 | - | - | - |
| Overarching urban paradigms | For pedestrians only | - | - | - | - | - | - | - | 3 | - | - | - | - | - | - | 1 | - |
| | Pedestrian-only streets | - | - | - | - | - | - | - | - | - | - | - | - | - | - | - | - |
| | Pedestrian-only zones | 5 | - | - | - | - | - | - | 1 | - | - | - | - | - | - | - | - |
| | Pedestrian priority streets | - | - | - | - | - | - | - | - | - | - | - | - | - | - | - | - |
| | Pedestrian-friendly (PF) | - | - | - | 1 | - | - | - | - | - | - | - | - | - | - | 4 | - |
| | PF footpaths | 1 | - | - | - | - | - | - | - | - | - | - | - | - | - | - | - |
| | PF environment | - | - | - | - | - | 1 | - | - | - | - | - | - | - | 1 | 1 | - |
| | PF built environments | - | - | - | - | - | - | - | - | - | - | - | - | - | 1 | - | - |
| | PF spaces | - | - | - | - | - | 1 | - | - | - | - | - | - | - | - | - | - |
| | Pedestrian-oriented (PO) | - | 2 | - | - | - | - | - | - | - | - | - | - | - | - | - | - |
| | PO streets | - | - | - | - | - | - | - | - | - | - | 1 | - | - | - | - | - |
| Methods or techniques | Pedestrian flow | - | - | - | - | - | - | - | - | - | - | - | - | - | - | - | - |
| | Pedestrian footfall | 7 | - | - | - | - | - | - | 1 | - | - | - | - | - | - | - | - |
| | PF design | - | - | - | - | - | - | - | - | - | - | - | - | - | - | - | - |
| | Pedestrian mobility | - | - | - | - | 1 | - | - | - | - | - | - | - | - | - | - | - |
| | Pedestrian behaviors | - | - | - | - | 45 | - | - | - | - | - | - | - | - | - | - | - |

**Table 3.** *Cont.*

| Notes | Near Synonym (Manuscripts *) | 1 | 2 | 3 | 4 | 5 | 6 | 7 | 8 | 9 | 10 | 11 | 12 | 13 | 14 | 15 | 16 |
|---|---|---|---|---|---|---|---|---|---|---|---|---|---|---|---|---|---|
| Tactical approaches | Pedestrian schemes | - | 2 | - | - | - | - | 4 | - | - | - | - | - | - | - | - | - |
| | Pedestrian mall strategies | - | - | - | 2 | - | - | - | - | - | - | - | - | - | - | - | - |
| | Pedestrian street design | - | - | - | 2 | - | - | - | - | - | - | - | - | - | - | - | - |
| | Pedestrian policies | - | - | - | - | - | 1 | - | - | - | - | - | - | - | - | - | - |
| | PF schemes | - | - | - | - | - | - | - | - | - | - | - | - | - | - | 1 | - |
| | Pedestrian priority scheme | - | - | - | - | - | - | - | - | - | - | - | - | - | - | - | - |
| | PO schemes | - | - | - | - | - | - | - | - | 1 | - | - | - | - | - | - | - |
| | Pedestrian zone schemes | - | - | - | - | - | - | - | - | 1 | - | - | - | - | - | - | - |
| Miscellaneous | Shopping centers | - | - | - | - | - | - | - | - | 3 | - | - | - | - | - | - | - |
| | Shopping streets | - | 5 | - | - | - | - | - | - | - | - | - | 3 | - | - | - | - |
| | Shopping malls | - | - | - | - | - | - | - | - | - | - | - | 1 | - | - | - | - |
| | Suburban shopping centers | - | - | - | - | - | - | - | - | 17 | - | - | - | - | - | - | - |

\* The first column includes the six factors of interest. 1. Allirani, Dumka, and Verma (2024) [3], 2. Feriel (2023) [21], 3. te Boveldt, Wilde, Keseru, and Macharis (2023) [39], 4. Gregg (2023) [22], 5. Yıldırım and Çelik (2023) [48], 6. Yoshimura, et al. (2022) [40], 7. Villani and Talamini (2021) [51], 8. Murakami, Villani, and Talamini (2021) [42], 9. Gregg (2019) [43], 10. Yassin (2019) [49]. 11. Parajuli and Pojani (2018) [32], 12. Özdemir and Selçuk (2017) [41], 13. Zainol et al. (2016) [44], 14. Nakamura (2016) [46], 15. Soni and Soni (2016) [45], and 16. Castillo-Manzano, Lopez-Valpuesta, and Asencio-Flores (2014) [47].

The urban planning and design literature reveals a consistent and nuanced use of near synonyms related to pedestrianization. Terms like "pedestrian", "pedestrian area/areas", "pedestrian zone/zones", "pedestrianized streets/areas/zones/environments", and "pedestrianization" appear across assorted studies. For example, "pedestrian" [3,21,46], "pedestrian area/areas" [21,45,48,49], and "pedestrian zone/zones" [3,21,42,47,48] are recurrent. Similarly, terms like "pedestrianized streets" [22,43,47,48] and "pedestrianized areas" [40,41,48] are prevalent. Despite this consistency, slight conceptual or terminological nuances exist among authors. Table 4 shows an overview of the numerical analysis of near-synonym concepts and terms. In Figure 4, the authors' employment of certain near synonyms in the noun case is illustrated, such as "pedestrian area" or "areas", "pedestrian street" or "streets", and "pedestrianized zone" or "zones", which were also predominantly utilized in the verb case, as seen with "pedestrianized area" or "areas" and "pedestrianized streets" or "streets". However, these near synonyms were not extensively employed in the case of the adjective "pedestrianizing".

**Table 4.** Overview of numerical analysis of near-synonym concepts and terms.

| Near Synonym (Manuscripts *) | 1 | 2 | 3 | 4 | 5 | 6 | 7 | 8 | 9 | 10 | 11 | 12 | 13 | 14 | 15 | 16 |
|---|---|---|---|---|---|---|---|---|---|---|---|---|---|---|---|---|
| Pedestrianize | - | - | - | - | - | - | - | - | - | - | - | - | - | - | 4 | - |
| Pedestrianization | 76 | 44 | 83 | 84 | 39 | 39 | 93 | 36 | 57 | 61 | 71 | 81 | 21 | 7 | 104 | 1 |
| Pedestrianization of an area | - | - | - | - | - | - | - | - | - | - | - | - | - | - | 2 | - |
| Pedestrianization schemes | - | - | 12 | - | - | - | - | - | 2 | 9 | 7 | 29 | - | - | - | 7 |
| Pedestrianization strategies | - | - | - | 16 | - | - | - | - | - | - | - | - | - | - | - | - |
| Pedestrianization of the central boulevards | - | - | 1 | - | - | - | - | - | - | - | - | - | - | - | - | - |
| Pedestrianization process | - | - | - | - | - | - | - | - | - | - | - | - | - | - | - | 14 |
| Post-war pedestrianization | - | - | - | 5 | - | - | - | - | - | - | - | - | - | - | - | - |
| Pedestrianization policy | - | - | - | - | 1 | 2 | - | - | - | - | - | - | - | - | - | - |
| Pre-pedestrianization | - | - | - | - | - | - | - | - | - | - | - | - | - | - | - | 1 |
| Street pedestrianization | - | - | - | - | 2 | 2 | - | - | - | - | - | - | - | - | - | - |
| Pedestrianized | 29 | 7 | - | - | 9 | - | 58 | 1 | 13 | 2 | 9 | 24 | - | 70 | 37 | 18 |
| Pedestrianized area | - | 2 | - | - | 17 | - | 15 | - | - | - | 2 | 24 | - | 2 | 6 | - |
| Pedestrianized center | - | - | 1 | - | - | - | - | - | - | - | - | - | - | - | - | - |
| Pedestrianized districts | - | - | - | - | - | - | - | - | - | - | 1 | - | - | - | - | - |

**Table 4.** *Cont.*

| Manuscripts * / Near Synonym | 1 | 2 | 3 | 4 | 5 | 6 | 7 | 8 | 9 | 10 | 11 | 12 | 13 | 14 | 15 | 16 |
|---|---|---|---|---|---|---|---|---|---|---|---|---|---|---|---|---|
| Pedestrianized place | - | - | - | - | 1 | 1 | - | - | - | - | - | - | - | - | - | - |
| Pedestrianized space | - | - | - | - | - | - | 11 | - | - | - | - | - | - | - | - | - |
| Pedestrianized streets | - | - | - | - | - | - | 10 | - | 5 | - | - | - | - | 36 | 1 | 8 |
| Pedestrianized square | - | - | - | - | - | - | - | - | - | - | - | - | - | - | - | - |
| Pedestrianized zone | - | - | 1 | - | - | - | - | - | - | - | - | - | - | - | - | 4 |
| Pedestrianized urban street | 5 | - | - | - | - | - | - | - | - | - | - | - | - | - | - | - |
| Pedestrianized city centers | - | - | - | - | - | - | - | - | - | - | 1 | - | - | - | - | - |
| Non-pedestrianized | - | - | - | - | - | - | - | - | - | - | - | - | - | 39 | - | - |
| Pedestrianizing | - | - | 5 | - | - | - | - | - | 5 | - | - | - | - | - | - | - |
| Pedestrianizing area | - | - | - | - | - | - | - | - | - | - | - | - | - | 1 | - | 1 |
| Pedestrianizing boulevard | - | - | 1 | - | - | - | - | - | - | - | - | - | - | - | - | - |
| Pedestrianizing main streets | - | - | - | - | - | - | - | - | - | - | - | - | - | - | - | - |
| Pedestrianizing streets | - | - | - | 2 | - | - | - | - | 2 | - | - | - | - | - | - | - |

\* These numbers indicate the same manuscripts in Table 3.

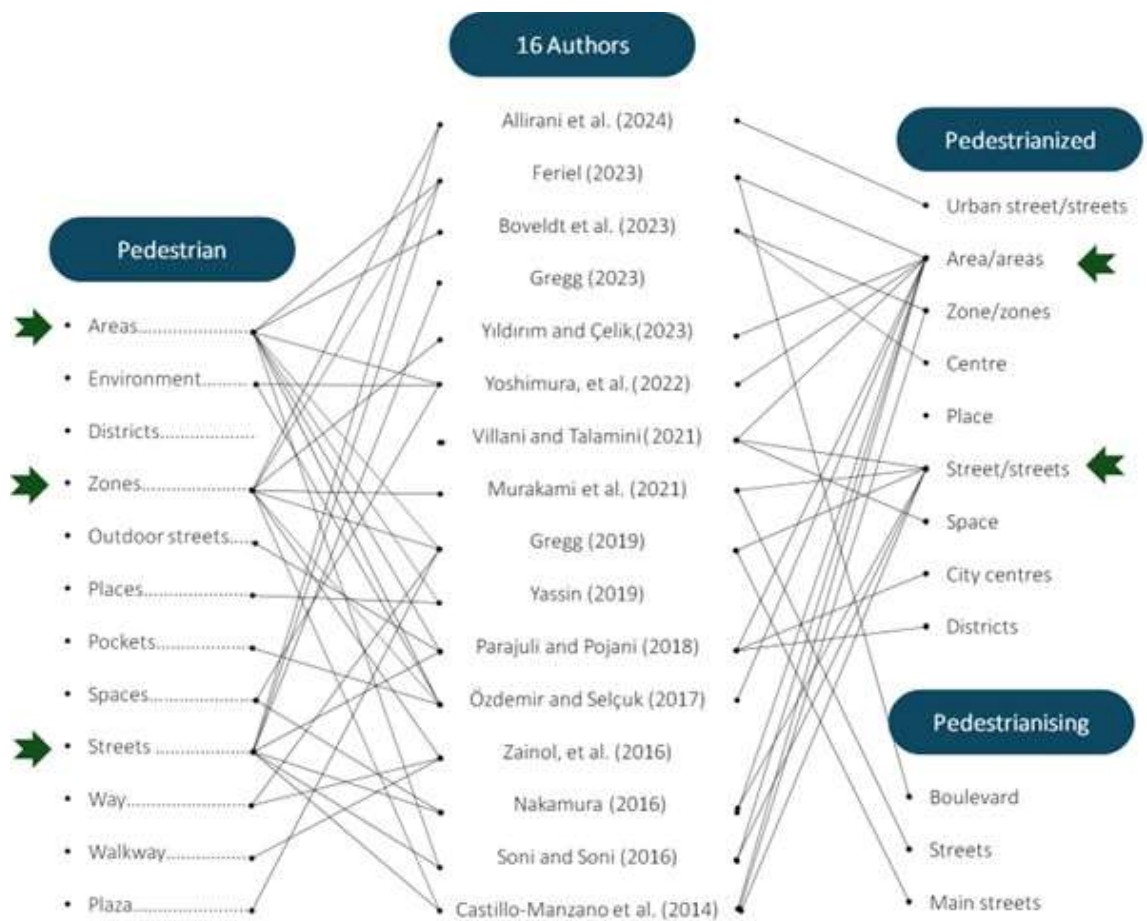

**Figure 4.** Common near synonyms associated with pedestrian. The dark green arrows indicate the interchangeable usage of synonyms within the analyzed text [3,21,22,32,39–50].

The lexicon about pedestrian-related discourse encompasses multifaceted dimensions, embracing analytical methods or techniques such as "pedestrian footfall" and "pedestrian street design", as well as actionable strategies like "pedestrian schemes", "pedestrian-friendly schemes", "pedestrian mall strategies", "pedestrian-oriented schemes", and "pedestrian priority schemes". It further encompasses overarching paradigms, emerging trends, and strategic approaches, epitomized by terms such as "pedestrian-friendly", "pedestrian-friendly footpaths", "pedestrian-friendly environments", "pedestrian-friendly

built environments", "pedestrian-friendly spaces", "pedestrian mobility", and "pedestrian-oriented cities".

Additionally, the attributes intrinsic to pedestrian dynamics, such as "pedestrian flow" and "pedestrian behavior", are also elucidated within this discourse, to shed light on the authors' utilization of near synonyms and their categorization based on six facets of interest, including elements of urban landscape design, overarching urban paradigms, methodologies, and actionable strategies.

Below, we provide a comprehensive overview of recurring near synonyms among authors, highlighting notable instances such as "pedestrian malls", "for pedestrian only", "pedestrian way", and "pedestrian friendly". Subsequently, we delineate the utilization of near-synonym terms or concepts conveying identical meanings. Lastly, we present examples of near synonyms introduced by individual authors for the first time (Figure 5).

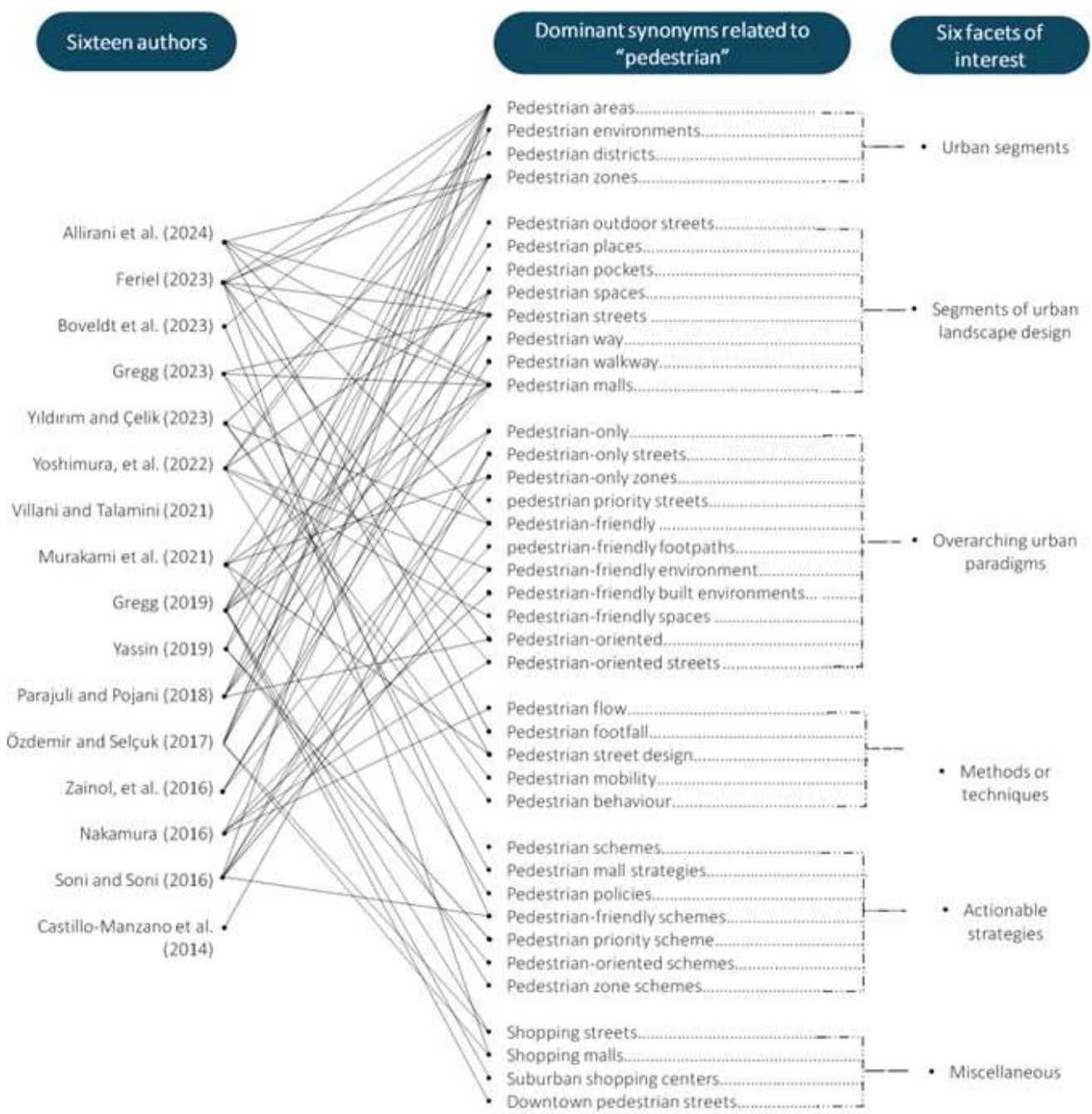

**Figure 5.** Common near synonyms associated with pedestrian and six facets of interest [3,21,22,32,39–50].

1.  Pedestrian Malls

    Parajuli and Pojani (2018) [32], Murakami et al. (2021) [42], Gregg (2019; 2023) [22,43], Feriel (2023) [21], and Allirani et al. (2024) [3] describe pedestrian malls as pedestrianized areas typically found in commercial areas or city centers, usually containing shops, restaurants, and other amenities. Gregg (2019) [43] says Gruen's ideas contributed significantly to the pedestrian mall movement, which gained popularity in the mid-20th century as cities sought to reduce urban congestion. Parajuli and Pojani (2018) [32] emphasize the association between pedestrian-only streets or malls and the quality of life, noting that pedestrian malls are considered places for retail activity in six European cities.

    Murakami et al. (2021) [42] mention the term "pedestrian mall" once in their text while discussing its commercial success in American cities in the 1980s due to inadequate parking space design and operation. Feriel (2023) [21] emphasizes that "pedestrianization" emerged to separate pedestrians from vehicular traffic in European city centers. He also stated that originating from the pioneering work of Victor Gruen, who introduced the concept of "pedestrian malls" to create pedestrian-friendly spaces in North American cities, pedestrianization has become a favored strategy for downtown retail revitalization [43].

    Concurrently, European city planners in the 1960s introduced the term "pedestrianization" to delineate efforts to separate pedestrians from vehicular traffic in downtown shopping areas. Allirani et al. (2024) [3] also referenced this term twice, attributing its emergence to Victor Gruen's work and noting a need for more documentation regarding this experience in European cities.

2.  For Pedestrians Only

    Soni and Soni (2016) [45], Murakami et al. (2021) [42], and Allirani et al. (2024) [3] commonly utilize the term "for pedestrians only" to denote a specific designation or restriction in street design, where access is exclusively limited to pedestrians, thereby prohibiting vehicular traffic. While this concept aligns with overarching urban paradigms, it is not necessarily classified as a movement or trend; instead, it represents a design approach or concept aimed at prioritizing pedestrian use and safety within urban environments. This approach resonates with movements advocating for pedestrian-friendly urban design and sustainable transportation solutions, contributing to the broader initiative aimed at creating walkable cities and promoting pedestrian-oriented urban spaces.

3.  Pedestrian Way

    The divergence in the usage and interpretation of the term "pedestrian way" by Zainol et al. (2016) [44] and Gregg (2019) [43] underscores the complexity inherent in the historical context and the evolution of concepts within urban design discourse. Zainol et al. (2016) [44] utilize the term "pedestrian way" to describe a designated area for pedestrian activity within the urban landscape, akin to a pedestrian street. However, they do not explicitly delve into its historical origins or conceptual foundations. In contrast, Gregg (2019) offers a more nuanced perspective, tracing the term back to its inception in the mid-20th century. She stated that "pedestrian way" emerged as a strategic response to downtown decline, particularly in areas formerly dominated by automobile-oriented infrastructure.

    This conceptualization involved the transformation of main thoroughfares into pedestrian-friendly zones, revitalizing urban cores and fostering pedestrian engagement. Gregg (2019) [22] further highlights the historical significance of the "pedestrian way" concept, citing its pivotal role in downtown redevelopment endeavors, such as those in Syracuse, NY. Architects Jo van den Broek and Jacob Bakema contributed to its refinement in the mid-20th century, while Arthur McVoy solidified its presence in urban planning discourse with proposals for a "pedestrian way" business district in Baltimore during the late 1950s.

4.  Pedestrian-Friendly

    The findings of this study highlight a recurring theme among multiple authors, namely the frequent utilization of the term "friendly" to characterize urban streets, underscoring a

commitment to enhancing convenience for pedestrians and non-motorized users. Nakamura (2016) [46] introduces the concept of "pedestrian-friendly environments", a notion echoed by Yoshimura et al. (2022) [40] and Özdemir and Selçuk (2017) [41] in their exploration of "pedestrian-friendly spaces". Similarly, Soni and Soni (2016) [45] delve into "pedestrian-friendly schemes", aligning with this overarching concept. In the subsequent discussion, we provide examples of near-synonym terms employed by various authors in elucidating these ideas.

Examining the economic implications of enhancing pedestrian environments, Soni and Soni (2016) [45] highlight Hasselt in Belgium. This Belgian city has transformed its streets into "pedestrian-friendly environments" to boost footfall, sales, and rent. Similarly, Yoshimura et al. (2022) [40] emphasize the importance of "pedestrian-friendly spaces" in improving the shopping experience and attracting more customers to retailers, increasing sales volume. In a similar vein, Soni and Soni (2016) [45] utilize various terms such as "pedestrian-friendly area" and "pedestrian-friendly schemes" to convey similar meanings, observing that pedestrian-friendly areas tend to attract more pedestrians and contribute to creating a pedestrian-friendly environment.

Nakamura (2016) [46] employs the terms "pedestrian-friendly environments" and "pedestrian-friendly built environments" interchangeably, stressing the importance of specific design elements in urban street design. Similarly, Yoshimura et al. (2022) [40] elaborate on pedestrianization as the transition from vehicular use to a "walkable environment", introducing near-synonym terms such as "pedestrian environments" and "non-pedestrian environments" within the context of creating walkable urban spaces. Additionally, they discuss "pedestrian-friendly environments" alongside "vehicle-oriented" concepts, illustrating the coexistence of pedestrian-centric and vehicle-centric perspectives.

Feriel (2023) [21] delves into "pedestrian schemes", whereas Yassin (2019) [49], Castillo-Manzano et al. (2014) [47], and Murakami et al. (2021) [42] discuss "pedestrianization schemes". Both concepts share the objective of improving pedestrian experiences and cultivating pedestrian-friendly urban environments. These initiatives prioritize pedestrian requirements over vehicular traffic, enhancing mobility and contributing to the creation of more sustainable and livable cities.

To illustrate the consistent usage and significance of the term "pedestrian-friendly" throughout the text, the authors employ it with the same emphasis as the elements and principles of street design and pedestrianization approaches. Yassin (2019) [49] emphasizes "pedestrian-friendly access" as a means to transition mobility from car-oriented to pedestrian-oriented. Murakami et al. (2021) [42] discuss "pedestrian-friendly design elements", which encompass features that mitigate traffic congestion, reduce physical risks, establish a sense of place, attract diverse demographics, foster community connectivity, and facilitate social interactions and spending at local establishments such as retail shops, workshops, galleries, cafes, and restaurants.

Similarly, Allirani et al. (2024) [3] introduce the term "pedestrian-friendly footpaths" to delineate street design principles, including granite cobblestone pavement, designated parking areas for both two-wheelers and four-wheelers and strategically placed dustbins. It is worth noting that not all authors utilize the term "friendly" in their discussions. Özdemir and Selçuk (2017) [41], Parajuli and Pojani (2018) [32], Yassin (2019) [49], Villani and Talamini (2021) [51], te Boveldt et al. (2023) [39], Feriel (2023) [21], Gregg (2019; 2023) [22,43], and Yıldırım and Çelik (2023) [48] are among those who do not employ the term "friendly".

- The second group: The near synonyms of pedestrianize

The second repetition of near synonyms correlated to the term "pedestrianize", which is utilized as "pedestrianization", "pedestrianization schemes", "pedestrianized", and "pedestrianizing". Each term appears in three cases: the broader concept or action of making, the state after being converted, and the active transformation or changes. It appears that Soni and Soni (2016) [45] employed the term "pedestrianize" as a verb to denote the process of transforming an area into a pedestrian zone. Moreover, they utilized

it in a broader context to underscore the advantages of pedestrianization, emphasizing enhancements in mobility and environmental quality. Likewise, Yassin (2019) [49] utilized the term "pedestrianize" in an overarching manner, indicating a concerted effort towards rendering an area more conducive to pedestrian activity and safety.

1. Pedestrianization

Over the past decade, scholars in urban planning and design have characterized pedestrianization in multifaceted ways. It has been conceptualized as a process, measure, device, key component, strategy, policy, approach, and pivotal step in societal transformation [21,32,39–41,43,45,47,48,50,51]. This rich array of conceptualizations underscores the complexity and significance of pedestrianization in urban development.

Many authors across the sixteen articles uniformly utilized the concept or term 'pedestrianization', which refers to the creation of spaces entirely closed to vehicles. Allirani et al. (2024) [3] define vehicles as cars, two-wheelers, and commercial vehicles. According to Feriel (2023) [21], pedestrianization occurs when streets are closed and renovated. The goal of pedestrianization is to enable pedestrians to move freely, combat car dominance, promote sustainable modes of transportation [39], and develop the economic sector through improvements in retail shopping [22,40,42,46,48]. Pedestrianization also helps to reduce negative environmental effects, such as air pollution and noise [54]. Additionally, pedestrianization can create a pleasant environment for pedestrians, providing opportunities for social interaction and leisure activities [40,45,49,55].

Pedestrianization often implies the allocation of space for pedestrians; however, scholars have delineated it further, distinguishing it from concepts like "traffic calming" and emphasizing its social, economic, perceptual, and environmental implications [32,41,47], and pedestrian malls were considered as one of the broader concepts of pedestrianization that was used to refer to pedestrian areas in central cities of the United States during World War II [22].

2. Pedestrianization Schemes

The term "scheme" is utilized among urban planners to denote the implementation process, with a shared understanding of its semantic implications. Meanwhile, various near synonyms were employed, such as "pedestrianization schemes" by Castillo-Manzano et al. (2014) [47], Soni and Soni (2016) [45], Özdemir and Selçuk (2017) [41], Parajuli and Pojani (2018) [32], Yassin (2019) [49], Murakami et al. (2021) [42], and te Boveldt et al. (2023) [39]. Soni and Soni (2016) [45] introduced "pedestrian-friendly schemes", while Yassin (2019) [49] and Murakami et al. (2021) [42] referred to "pedestrian-oriented schemes" and "pedestrian priority schemes", respectively. Feriel (2023) [21] opted for "pedestrian schemes".

In discussions concerning historical preservation, Soni and Soni (2016) [45] delineated 'pedestrianization schemes' to encompass methods, costs, and sustainability aspects. Conversely, they associated 'pedestrianization' with negative impacts such as vehicular pollutants, vibrations, encroachment, and maintenance issues. Parajuli and Pojani (2018) [32] and Yassin (2019) [49] highlighted regional disparities in pedestrianization schemes across European, North American, Australian, Asian, and North African cities. European schemes focus on traffic regulation, pollution control, architectural conservation, environmental enhancement, and retail revitalization.

Conversely, North American initiatives prioritize downtown revitalization and pedestrian orientation without complete car elimination, while Australian efforts emphasize land use and transportation patterns. Asian schemes aim to reduce car dependency, enhance pedestrian safety, and stimulate economic activity, while North African schemes target urban fabric revitalization.

Yassin (2019) [49] says pedestrianization schemes are integral to creating high-quality spaces and contributing to livable environments. However, more than designating an area as a pedestrian zone alone is required for livability; streets must also serve multiple functions, including facilitating socialization, community interaction, and economic activity. Yassin (2019) [49] expands this concept to encompass four overarching goals: regulating

vehicular access, preserving historic architecture, revitalizing downtown areas, improving land use patterns, and enhancing pedestrian safety and convenience.

3. Pedestrianized

It seems that the term "pedestrianized" was commonly used by many authors, but the specific usage to describe a place varied among them. While most authors employed phrases like "pedestrianized area" "pedestrianized place", "pedestrianized zone", and "pedestrianized center", Yassin (2019) [49], Zainol (2016) [44], and Murakami (2021) [42] did not utilize these near synonyms. Instead, they described the area or zone using "pedestrian".

Castillo-Manzano et al. (2014) [47], Nakamura et al. (2016) [46], and Feriel (2023) [21] frequently employed the term "pedestrianized area/areas" in their studies, while Castillo-Manzano et al. (2014) [47] and Nakamura et al. (2016) [46] opted for "pedestrianized streets" instead. In contrast, Soni and Soni (2016) [45] utilized "pedestrianized streets" once and used "pedestrianized areas" thereafter. Similarly, Villani and Talamini (2021) [51] consistently used these terms and added "pedestrian spaces". Yoshimura et al. (2022) [40] and Yıldırım and Çelik (2023) [48] introduced "pedestrianized places" once in their analyses. Allirani et al. (2024) [3] specified "pedestrianized urban streets/streets", while te Boveldt et al. (2023) [39] referred to "pedestrianized zones" and "pedestrianized centers" once each.

4. Pedestrianizing

Castillo-Manzano et al. (2014) [47] referred to "pedestrianizing areas" only once, depicting the state of older areas within the city. They used terms such as "pedestrianized street/streets" and "pedestrianized zones" to denote streets and areas inhabited by pedestrians. In their examination of social benefits, Soni and Soni (2016) [45] employed the term "pedestrian areas" to denote the positive effects of pedestrianization. However, when discussing social interaction, relations, and the reduction in air pollution in the context of environmental benefits, they utilized the term "pedestrianized streets", consistent with other authors. In addition, Nakamura (2016) [46] included pedestrianized squares, courts, and single pedestrianized streets under "pedestrian areas".

- The third group: Indirectly associated near synonyms

The third group encompasses three distinct types of indirectly associated near synonyms. Firstly, it incorporates concepts or terms emblematic of an alternative intellectual paradigm, exemplified by "walkability", "walkability-friendly environment", "walkable built environments", "walkable city" or "cities", "walkable street", "walkable street corridor", and "walking environment". Secondly, it addresses the dichotomy between pedestrians and vehicular traffic, encapsulated by terms like "car-oriented", "car-oriented city", "car-free", "car-free zones", "car-free urban living", "car-dependent suburbia", "car-free boulevard", "traffic-free areas", "vehicle-oriented", and "traffic-calming".

This group encompasses near synonyms of independent concepts facilitating urban pedestrian movement, such as "boulevards", "complete streets", "shopping centers", "shopping malls", "shopping streets", and "skywalk networks". The study focuses solely on the third type, while the first and second lead to different research directions.

1. Boulevards

Similarly, te Boveldt et al. (2023) [39] are noteworthy for being the only researchers to mention the concept of boulevards. They referenced this concept when discussing the pedestrianization scheme for Rue Neuve, a prominent shopping street in the renowned Grand Place (Ilˆot sacré) area in the Anspach district of Brussels. The authors expressly referred to the central boulevards of Brussels, where Rue Neuve is located, as boulevards.

2. Complete Streets

Complete streets share concepts such as "pedestrianization", "streets for pedestrians only", "streets for people", and "walkable streets", emphasizing the importance of accommodating all forms of transportation. In this work, Gregg (2023) [22] stands out as the

sole researcher to mention the concept of complete streets. She defines this concept as a thoroughfare accommodating automobiles, bicycles, and pedestrians within the street right-of-way. She emphasizes that this concept superseded the pedestrian mall design proposed by Gruen and Landscape Architect Garrett Eckbo.

3.    Shopping Centers

Gregg (2019) [43] argues that "shopping centers" epitomize a wider concept within pedestrianization, tracing their origins to North American suburbs engineered for pedestrian use. As a result, she posits that European pedestrianization initiatives were shaped by the planned shopping center model originating in North America. For instance, Gregg (2023) [22] highlights the Lijnbaan pedestrian mall in Rotterdam, Netherlands, inaugurated in 1953, as a prime example of post-war urban development embodying pedestrianization ideals in Europe. This cross-pollination of ideas underscores the reciprocal influence between Europe and North America on urban design paradigms.

4.    Shopping Malls

Özdemir and Selçuk (2017) [41] stand out as the only researchers who employed the term 'shopping malls', which carries a similar meaning to 'pedestrian malls'. They noted that shopping streets emerged from European planner's efforts to revitalize areas such as Lijnbaan Street in Rotterdam and other cities in West Germany, particularly in the aftermath of war damage, during the 1950s.

5.    Shopping Streets

Özdemir and Selçuk (2017) [41] and Feriel (2023) [21] employed the term 'shopping streets' to refer to traditional urban streets designated for shopping in city centers and historic districts. Feriel (2023) [21] notably distinguished between shopping centers and pedestrianized areas.

6.    Skywalk Networks

Murakami et al. (2021) [42] are the only researchers to discuss skywalk networks, citing an example from Kowloon in Hong Kong. This pattern falls under the pedestrian zone plans classified within the vertical urbanism approach. The objective is to establish suspended corridors above the city center for retail trade. However, this approach presents two conflicting aspects: on the one hand, the competitive advantage of commercial areas in attracting international investments, companies, and tourists; on the other hand, the negative impact of segregating affluent office workers from street economies, potentially creating a perception of commercial areas as elitist.

4.3.2. Similar Terms Conveying Divergent Meanings

In the discourse surrounding pedestrianization, Soni and Soni (2016) [45] advocate for the comprehensive development of "pedestrianization schemes" and policies aimed at fostering pedestrian-friendly transportation systems, non-motorized transportation options, transit-friendly infrastructure, and supportive land use patterns. This approach underscores the multifaceted nature of pedestrianization initiatives, emphasizing the need for integrated strategies to promote pedestrian mobility and urban sustainability.

From an economic standpoint, Parajuli and Pojani (2018) [32] and Gregg (2019) [43] used the term "suburban shopping centers", referring to "pedestrianization", as a strategy to ensure the financial viability of retail establishments. The authors highlight the transformation of city centers into "pedestrian-friendly spaces" as a means to enhance the attractiveness of these areas and promote economic development. They underscore pedestrianization as a critical component of broader urban development projects aimed at revitalizing city centers and creating vibrant, livable communities.

A notable aspect observed in the discussion is the use of seemingly similar terms that convey divergent meanings. For instance, Yoshimura et al. (2022) [40] explore a spectrum of concepts and terms, including "pedestrian spaces", "environments", and "areas", to

elucidate the essence of pedestrianization. While their primary focus is on distinguishing between pedestrian and non-pedestrian environments, they introduce additional terms like "pedestrian-friendly environments" and "spaces" without clear justification.

Furthermore, they introduce innovative terminology such as "pedestrian grids" and "non-pedestrian grids" to analyze variations in in-store sales volumes, reflecting a nuanced understanding of pedestrian dynamics within urban spaces.

Moreover, Yoshimura et al. (2022) [40] delve into the concept of "pedestrian streets" to investigate shopping experiences, particularly those associated with purposeful and destination-driven trips. By highlighting the integral role of these experiences in urban vibrancy and the promotion of urban agglomeration, they underscore the interconnectedness between pedestrianization efforts and the overall vitality of urban environments. This discussion prompts further exploration into the diverse range of concepts and terms employed in the discourse on pedestrianization, reflecting the evolving nature of urban planning and design strategies aimed at enhancing pedestrian mobility and livability.

### 4.3.3. The Juxtaposition of Unrelated Vocabulary Lacking Semantic Resemblance

The juxtaposition of unrelated vocabulary, devoid of semantic resemblance, is a notable phenomenon observed in the discourse on pedestrianization. For instance, terms like "transport schemes" and "pedestrianization" or "main street" as a "pedestrian way" highlight this lexical diversity. Through a qualitative content analysis-based summative approach applied across sixteen selected articles, this study unveils intriguing insights into the semantic relationships and nuances inherent in the vocabulary under scrutiny, shedding light on the concept of "pedestrianization".

Soni and Soni (2016) [45] introduce terms like "pedestrians only" and "car-free", which signify the conversion of areas into pedestrian zones devoid of motor vehicles. They further elucidate the notion through terms like "pedestrianization scheme" and "pedestrianized area", offering a comprehensive understanding of the concept and its outcomes. Delving deeper into the objectives of pedestrianization, Özdemir and Selçuk (2017) [41] explore the creation of "car-free environments" conducive to "pedestrian-friendly shopping areas". They cite examples such as the revitalization of Lijnbaan Street in Rotterdam in the Netherlands, emphasizing the dual benefits of reducing traffic congestion and pollution while enhancing urban appeal. "Friendly" encompasses broader urban design goals to foster vibrant and livable streetscapes.

Similarly, te Boveldt et al. (2023) discuss "pedestrianization schemes" as transformative strategies adopted in Brussels, Belgium. Their focus extends beyond facilitating shopping, as seen in the 1950s, to strategically introducing car-free or low-traffic zones across urban areas. This contemporary approach aims to combat climate change by reducing car dominance in public spaces and enhancing the overall urban environment.

After a thorough examination of near-synonym concepts or terms utilized in pedestrianization research, it becomes imperative to devise a systematic approach to address the challenges associated with synonym selection. Therefore, we introduce the near synonym selection framework, which offers practical solutions to enhance clarity, precision, and effectiveness in scholarly communication within urban planning and design.

## 5. Discussion

The review above indicates that there is a reliable theoretical background, and the usage patterns of synonym terms in the pedestrianization literature reflect the multidimensional nature of urban planning and urban design and the complex interplay between economical, functional, social, cultural, and environmental dimensions. While the proliferation of synonymous terms may pose challenges to standardization and clarity, it also underscores the dynamic and evolving nature of academic pedestrianization discourse. By embracing this diversity of terminology, researchers and practitioners can enrich their understanding of pedestrianization and contribute to more holistic and inclusive approaches to urban planning and urban design.

*5.1. A Near Synonym Toolkit*

This section elucidates the components of the near synonym toolkit in the social sciences literature, which serves as a comprehensive guide to enhance the clarity and precision of near-synonym concepts and terms in the pedestrianization literature. The toolkit comprises three main components, organized logically to facilitate the seamless integration of near synonyms into research papers. These components include preparation and planning, synonym selection and integration, and refinement and evaluation. Each step within the toolkit is meticulously designed to optimize the utilization of near synonyms, ensuring coherence and effectiveness in scholarly discourse. The subsequent discussion delineates the ten key steps of the near synonym toolkit, drawing on insights from previous studies to provide practical guidance for researchers and practitioners navigating the complexities of near-synonym terms and concepts (Figure 6).

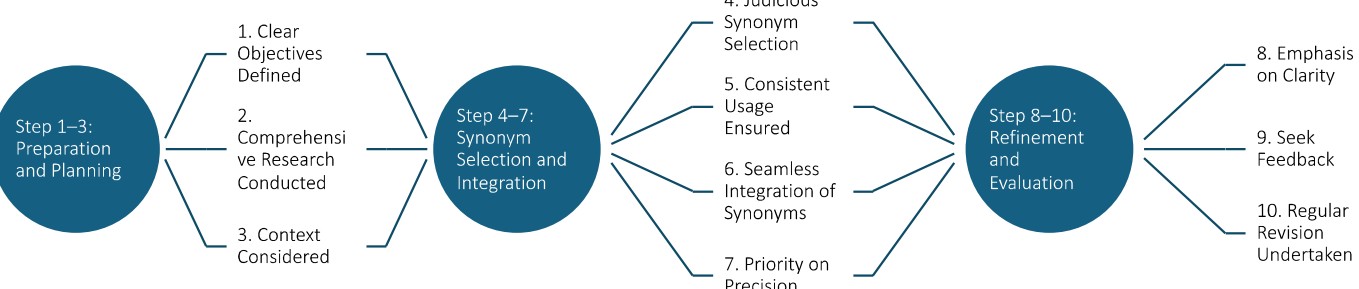

**Figure 6.** Three groups and ten steps of near-synonym toolkit.

These ten steps of the near-synonym toolkit are consistent with prior studies examining the unraveling of conceptual or terminological synonymy. Drawing on insights from these previous studies, decoding the lexical maze and unraveling conceptual or terminological synonymy in academic research can benefit from utilizing these ten steps as an initial near-synonym toolkit as follows:

1. Before selecting near synonyms, it is crucial to clearly define the primary aims and objectives of the research paper. This involves identifying the key concepts and terms central to the study and ensuring that the use of near synonyms aligns with these objectives [5].
2. A comprehensive literature review is performed to identify existing near synonyms and their usage in relevant scholarly works [56].
3. The context in which near synonyms are employed is meticulously deliberated, considering the particular field of study, scope, target audience, and semantic preferences [14,57].
4. Ensure that all near synonyms of concepts and terms are selected based on their appropriateness, relevance, and alignment with the defined aims and objectives of the academic research article. It is advisable to utilize the criteria for distinguishing near synonyms as outlined by Petcharat and Phoocharoensil (2017) [6] in their article.
5. Consistently employ chosen near synonyms to mitigate confusion and improve readability, a practice also advocated by Petcharat and Phoocharoensil (2017) [6].
6. The selected near synonyms are smoothly integrated into the text, ensuring they fit naturally within the narrative flow, which improves communication ideas.
7. Near synonyms are utilized with precision to convey intended meanings accurately and effectively [58].
8. Employing near-synonym expressions aims to improve readers' clarity and understanding, particularly by elucidating complex concepts and terms within their natural context [4].
9. Peer feedback is sought to evaluate the effectiveness of synonym usage and make necessary adjustments [59].

10. Periodic revision ensures consistency and clarity in writing [60].

The toolkit mentioned above provides structured guidance to researchers in selecting near synonyms when refining their manuscripts. The toolkit includes criteria for near synonym selection, methodologies for assessing appropriateness, guidelines for ensuring consistency, and strategies for identifying relevant near synonyms within the urban planning and design domain. It is important to note that this article focuses specifically on loose or near synonyms that have looser similarities in sense and lack absolute meaning.

*5.2. A Framework for Near Synonym Selection*

The framework suggested in this section can be structured in three phases and thirteen steps (Table 5). In the first phase, researchers are guided through four steps to effectively utilize near synonyms in scientific research. This phase emphasizes the importance of clarity and precision in language usage. The second phase involves five steps to investigate near synonym gaps, termed 'semantic preference', within research. Researchers are encouraged to explore areas where near synonyms may be lacking or inconsistently applied, fostering a deeper understanding of terminology usage. Finally, the third phase offers four steps for collaborative efforts between researchers and practitioners in leveraging near synonyms effectively. By working together, stakeholders can ensure consistency and coherence in terminology usage across various contexts.

**Table 5.** Three phases and 13 steps of near synonym selection framework.

| Phase One (1–4) | Phase Two (5–9) | Phase Three (10–13) |
|---|---|---|
| Effective Utilization in Scientific Research | Investigating Research Gaps and 'Semantic Preference' | Consolidation of Near Synonym Usage |
| 1. Clarity and precision<br>2. Reduced confusion<br>3. Streamlined communication<br>4. Enhanced comparability | 5. Establish near synonym guidelines<br>6. Collaborative efforts<br>7. Near synonym mapping<br>8. Education and training<br>9. Regular review and update | 10. Synonym usage patterns<br>11. Cross-cultural perspectives<br>12. Effectiveness of stabilization efforts<br>13. Integration of semantic technologies |

5.2.1. Phase One (Steps 1–4): Effective Utilization in Scientific Research

In this phase, four steps elucidate the importance of consistent near synonyms:

1. Consistency in near synonyms enhances clarity and precision in communication, ensuring a uniform understanding of concepts across diverse contexts.
2. Avoiding unnecessary near synonyms reduces reader confusion, particularly among individuals from varied cultural and linguistic backgrounds, thereby facilitating a better comprehension of research findings.
3. The adoption of standardized terms streamlines communication within the field, fostering collaboration and knowledge exchange among researchers.
4. Consistent terminology enables easier comparison between studies, a crucial aspect for advancing knowledge and understanding within the discipline.

5.2.2. Phase Two (Steps 5–9): Investigating Research Gaps and 'Semantic Preference'

The second phase involves addressing gaps in synonym usage through five steps:

5. The development of guidelines or standards for terminology usage within the field, specifying preferred terms and discouraging unnecessary near synonyms.
6. Promoting collaboration among researchers, practitioners, and stakeholders to establish a shared terminology that reflects the diverse perspectives within the field.
7. Identifying prevalent near synonyms and usage patterns through systematic reviews or near synonym mapping exercises.
8. The provision of education and training programs to researchers, students, and practitioners, emphasizing the importance of consistent terminology usage and providing strategies for achieving it.

9.  The continuous review and updating of terminological guidelines to accommodate evolving concepts and emerging trends in urban planning and design.

5.2.3. Phase Three (Steps 10–13): Consolidation of Near Synonym Usage

The four steps aimed at understanding and consolidating near synonym usage are as follows:

10. An examination of factors influencing the choice of near synonyms in the research literature and their impact on communication and knowledge transfer.
11. Studying how cultural differences influence the interpretation and use of near synonyms within urban planning and design research, in international contexts.
12. An assessment of the effectiveness of initiatives to standardize terminology usage within the field and their implications for research quality and practice.
13. An investigation of the potential of semantic technologies, such as ontologies and controlled vocabularies, to facilitate consistent and interoperable terminology usage in urban planning and design research.

*5.3. Limitations*

It is crucial to acknowledge the limitations of this study. Firstly, it encompasses only a selection of near synonyms utilized in pedestrianization research, potentially excluding other variations. Moreover, the analysis relies on a limited number of articles and subjective interpretations of near-synonym concepts and terms, potentially overlooking insights from the non-English literature and trends in conceptual and terminology usage.

The second limitation pertains to the quantitative numerical analysis, conducted manually using the "find text" or "find all" commands on a keyboard. While effective for the sixteen articles analyzed in this study, this method may prove less efficient for larger datasets. We suggest employing advanced text mining and natural language processing (NLP) techniques for larger volumes of data. These techniques enable the identification of patterns, relationships, and near-synonym terms across a broad spectrum of the literature, automating the analysis process and saving time.

*5.4. Casual Argument, Speculations, and Deductive Arguments*

There exists a causal relationship between the meticulous selection and cohesive integration of near-synonym concepts and terms in pedestrianization research and the overall writing style of academic research. Speculative discussions propose that the emergence of novel terminology and concepts in pedestrianization research mirrors the evolving priorities and challenges within the realm of urban development.

Deductively, the interchangeable use of concepts and terms in pedestrianization discourse suggests a shared understanding of desired outcomes, reflecting a collective commitment to textual coherence rooted in established principles of urban planning and urban design. By analyzing near-synonym concepts and terms in pedestrianization, valuable insights into underlying motivations and implications in urban planning and design discourse can be gained. This approach offers an opportunity to enrich the inclusivity of the academic literature by elucidating the nuanced relationships between terminology and conceptual frameworks.

**6. Conclusions**

This study underscores the significance of addressing challenges related to near synonym usage in pedestrianization research. The complexities surrounding the conceptual and terminological near synonyms used in delineating pedestrianization dimensions have become increasingly apparent throughout the literature. Beginning by emphasizing the crucial role of terminological consistency in academic research, particularly in a specialized field like pedestrianization, this study discusses consistent terminology. Our argument for the consistency of near-synonym terms and concepts is to facilitate clear communication, enhance understanding, and foster collaboration among researchers, practitioners,

policymakers, and other stakeholders. These near synonyms encompass diverse factors of interest, from specific urban segments to overarching urban paradigms, including methods and techniques and tactical approaches, adding complexity to understanding trends and methodologies in the field.

By shedding light on the interchangeable nature of specific concepts or terms in pedestrianization discourse, this article aims to identify near synonyms and elucidate the implications of their inconsistent usage in the academic literature. While strict equivalence between terms is not enforced, discerning usage patterns is essential for maintaining clarity, coherence, and precision in scholarly discourse.

Effective communication and comprehension in pedestrianization research and practice hinge on addressing near synonym usage challenges. Two key research questions emerge: firstly, exploring strategies to promote the effective and consistent use of near synonyms, and secondly, delving into the importance of maintaining terminology consistency for enhancing communication and understanding within pedestrian studies.

Although near synonyms may appear similar and convey overlapping ideas, they serve distinct purposes in narrative and analysis, offering diverse perspectives on pedestrianization's impact and implications for urban planning and design. Embracing this diversity enriches our understanding of the subject, facilitating nuanced examinations of pedestrian-friendly urban environments across various contexts and highlighting the topic's complex, multifaceted nature.

To tackle the problem of using multiple near synonyms of a concept or term in a scientific manuscript, this study proposes a 10-step toolkit and structured framework comprising three phases and 13 steps for selecting near synonyms. In the first phase, the focus is on effectively utilizing near synonyms in scientific research. The second phase involves investigating research gaps to identify areas that need clarity. The third phase focuses on consolidating the usage of near synonyms and establishing 'semantic preference'. By following this structured approach, researchers can systematically tackle the challenges associated with the multiple uses of near synonyms in scientific writing. This enhances clarity, coherence, and consistency in communication within the scientific community.

Our suggested framework contributes to better communication for early-career researchers in urban planning and design when handling challenges related to pedestrianization research. By addressing these issues, the conceptual and terminological aspects of pedestrian research writing are broadened and might have an inaccurate use of near synonym terms or concepts, thereby offering crucial support for practitioners and academics engaged in urban planning and design.

This study's numerical analysis reveals a need for foundations, criteria, or rationales for selecting near synonyms in scientific research. Despite the limited sample size of sixteen studies, the findings underscore the urgent need to establish standardized methods for selecting, using, and repeating near synonyms in the academic literature. This issue poses significant challenges to scientific communication, research reproducibility, and pedestrianization research advancement.

Future research should expand the scope to encompass a broader range of the literature and languages. It should conduct longitudinal studies to track terminology changes and explore usage variations across regions and cultures. Additionally, insights from urban planners, policymakers, community members, and experts from related disciplines can further enrich our understanding of how near-synonym terms are perceived and utilized in pedestrianization discourse. Developing more sophisticated analytical methods to examine a large corpus of research articles and present the results in statistically robust and visually comprehensible ways is imperative. Future research might include exploring the impact of synonym use on the public understanding of pedestrianization concepts or developing software tools to assist in synonym selection. By addressing these challenges, researchers can ensure greater clarity, precision, and coherence in scholarly discourse on urban planning and design.

**Supplementary Materials:** The following supporting information can be downloaded at: https://doi.org/10.6084/m9.figshare.25378888.v3, File S1: Summary of 16 selected articles in 11 journals between 2014–2024.

**Author Contributions:** Conceptualization, H.A. and A.E.; methodology, A.E.; validation, H.A. and A.E.; formal analysis, H.A.; investigation, A.E.; resources, H.A.; data curation, A.E.; writing—original draft preparation, A.E. and H.A.; writing—review and editing, A.E.; visualization, A.E. All authors have read and agreed to the published version of the manuscript.

**Funding:** This research received no funding.

**Institutional Review Board Statement:** No research involving animals exists.

**Informed Consent Statement:** There are no animals involved.

**Data Availability Statement:** There is no information/issue to be collected.

**Acknowledgments:** It is with great appreciation that the authors would like to thank the editors and reviewers for their constructive comments.

**Conflicts of Interest:** The present study's authors declare that they have no conflicts of interest.

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
