# Peer review of "Decoding Near Synonyms in Pedestrianization Research: A Numerical Analysis and Summative Approach"

_urbansci, doi:10.3390/urbansci8020045_

Round 1

Reviewer 1 Report

Comments and Suggestions for Authors

The paper explores the intricate field of pedestranization research and examines the inconsistent use of almost synonymous words and concepts. This study aims to clarify linguistic difficulties in a term using a literary analysis, although the sample is not exhaustive. By emphasising the answer to the first research question - What toolkit ensures the effective and consistent use of near-synonymous terms in pedestrianship research? - the section on toolkits is not very clear. It is essential to clarify whether the discussion pertains to an established toolkit or a suggested toolbox for analysis. A concise summary of the toolkits for managing nearly identical terms would be more thorough. This section may contain examples of theoretical tools, lexical databases, and textual analysis software that have been proven helpful in past research. An in-depth review of the benefits and drawbacks of these tools would help establish a strong foundation for comprehending the context in which the new framework operates by clearly identifying the tool or tools utilised or created for analysis. When new tools are developed, it is essential to thoroughly explain their structure, use, and how they support the consistent and precise usage of almost identical terms.

To address the second research question effectively - Why is maintaining consistency important, and how can synonymous terms be chosen carefully? - the conclusions should highlight the significance of terminological consistency in pedestrianisation research, such as discussing how inconsistent term usage can confuse.

The conclusions should provide explicit instructions on choosing near-synonymous terms using the suggested toolbox based on fundamental concepts, including contextual relevance, conceptual precision, and alignment with current literature. Explaining the application of these concepts in the analysis can offer other researchers a reproducible framework for different situations.

The conclusions should be broadened to discuss how the work contributes to the academic community's discussion on pedestrianisation, its impact on future practices, potential research directions, the toolkit's relevance in other disciplines, and how consistent terminology can enhance interdisciplinary collaborations.

The paper could enhance the clarity of the research questions and provide a more comprehensive and forward-looking conclusion by implementing these modifications, highlighting the significance and influence of its contribution to research.

Author Response

The paper explores the intricate field of pedestranization research and examines the inconsistent use of almost synonymous words and concepts. This study aims to clarify linguistic difficulties in a term using a literary analysis, although the sample is not exhaustive.

Reply: Thank you for your valuable comment. Indeed, the sample size is small. This is one of our research limitations. Kindly check the sentences starts with “Moreover, the analysis relies on a limited number of articles …”   We have also added future research to cover this research limitation. Kindly check “Future research should expand the scope to encompass …”

By emphasising the answer to the first research question - What toolkit ensures the effective and consistent use of near-synonymous terms in pedestrianship research? - the section on toolkits is not very clear. It is essential to clarify whether the discussion pertains to an established toolkit or a suggested toolbox for analysis.

Reply: Thank you for your valuable feedback on our research section discussing toolkits in pedestrianization research. We appreciate your insights and the opportunity to clarify the intent of this section. We have revised this section to be included in the discussion section.

A concise summary of the toolkits for managing nearly identical terms would be more thorough. This section may contain examples of theoretical tools, lexical databases, and textual analysis software that have been proven helpful in past research. An in-depth review of the benefits and drawbacks of these tools would help establish a strong foundation for comprehending the context in which the new framework operates by clearly identifying the tool or tools utilised or created for analysis. When new tools are developed, it is essential to thoroughly explain their structure, use, and how they support the consistent and precise usage of almost identical terms.

Reply: Thank you for your valuable feedback on our research section discussing toolkits in pedestrianization research. We appreciate your insights and the opportunity to clarify the intent of this section. We have revised this section to be included in the discussion section.

To address the second research question effectively - Why is maintaining consistency important, and how can synonymous terms be chosen carefully? - the conclusions should highlight the significance of terminological consistency in pedestrianisation research, such as discussing how inconsistent term usage can confuse.

Reply: Thank you so much for your comment. We have added couple of sentences in the conclusion. Kindly check the paragraph that hold blue text start with “The complexities surrounding the conceptual and terminological near synonyms used in delineating pedestrianization dimensions have become increasingly apparent throughout the literature …”

The conclusions should provide explicit instructions on choosing near-synonymous terms using the suggested toolbox based on fundamental concepts, including contextual relevance, conceptual precision, and alignment with current literature. Explaining the application of these concepts in the analysis can offer other researchers a reproducible framework for different situations.

Reply: Thank you for this comment. We have added couple of sentences in the conclusion section. Please check the sentence starts with “To tackle the problem of using multiple near synonyms of a concept or term in a scientific manuscript, this study proposes a structured framework comprising three phases and 13 steps …”

The conclusions should be broadened to discuss how the work contributes to the academic community's discussion on pedestrianisation, its impact on future practices, potential research directions, the toolkit's relevance in other disciplines, and how consistent terminology can enhance interdisciplinary collaborations.

Reply: Thank you so much for your comment. We have added this sentences that start with “Our suggested framework contributes to better…”

The paper could enhance the clarity of the research questions and provide a more comprehensive and forward-looking conclusion by implementing these modifications, highlighting the significance and influence of its contribution to research.

Reply: We have made the necessary revisions based on your suggestions and have also included additional information to further enhance the content. Your input has been incredibly helpful in improving the overall quality of the manuscript. Please let us know if there are any other areas that you think require attention. Thank you again for taking the time to provide us with your feedback.

Reviewer 2 Report

Comments and Suggestions for Authors

While the introduction outlines a significant issue related to the use of near-synonyms in pedestrianization research, it would be beneficial to more explicitly state the research gap. Highlighting specific examples of confusion or misinterpretation in existing literature could strengthen the justification for this study.

The methodology section provides a good overview of the approach but could benefit from more detailed explanations of the criteria for selecting articles and journals. Clarifying how these criteria align with the study's objectives could enhance the reader's understanding of the research design.

There are instances where terms are used interchangeably without clear distinction (e.g., pedestrian areas, zones, environments). For clarity, define each term used and maintain consistency throughout the document, or explicitly discuss the rationale for using certain terms in specific contexts.

Lastly, suggesting specific areas for future research based on the findings could help guide subsequent studies. This might include exploring the impact of synonym use on public understanding of pedestrianization concepts or developing software tools to assist in synonym selection.

Author Response

While the introduction outlines a significant issue related to the use of near-synonyms in pedestrianization research, it would be beneficial to more explicitly state the research gap. Highlighting specific examples of confusion or misinterpretation in existing literature could strengthen the justification for this study.

 Reply: Thank you for your feedback. We have taken your comment seriously and incorporated the following sentence based on your guidance. Kindly check the paragraph that start with “The research identified a gap wherein pedestrianization …” and the example has been added “Sometimes, for example, some researchers use …”  and “However, despite the well-established…”

The methodology section provides a good overview of the approach but could benefit from more detailed explanations of the criteria for selecting articles and journals. Clarifying how these criteria align with the study's objectives could enhance the reader's understanding of the research design.

Reply: Thank you for bringing up this point. We have incorporated the following sentences to describe the criteria for selection. Kindly check the sentences that starts with “We curated a selection of sixteen peer-reviewed articles from eleven esteemed scientific journals relevant to our research do ...”

There are instances where terms are used interchangeably without clear distinction (e.g., pedestrian areas, zones, environments). For clarity, define each term used and maintain consistency throughout the document, or explicitly discuss the rationale for using certain terms in specific contexts.

Reply: We totally agreed with you. We have modified the sentences in the introduction section to provide an overview about the challenges of over usage of near synonyms in a signal manuscript. We have also provided a description of each near synonyms of ‘predestination.’   

Lastly, suggesting specific areas for future research based on the findings could help guide subsequent studies. This might include exploring the impact of synonym use on public understanding of pedestrianization concepts or developing software tools to assist in synonym selection.

Reply: Thank you for this comment. We have added a paragraph that describe the future research. Kindly check the sentences that starts with “Developing more sophisticated analytical methods to examine a large corpus of research articles and present the results in …”  

Round 2

Reviewer 2 Report

Comments and Suggestions for Authors

It can be accepted.